# Sucrose intensity coding and decision-making in rat gustatory cortices

Esmeralda Fonseca[1], Victor de Lafuente[2], Sidney A Simon[3], Ranier Gutierrez[1]*

[1]Laboratory of Neurobiology of Appetite, Department of Pharmacology, Center for Research and Advanced Studies of the National Polytechnic Institute, Mexico City, Mexico; [2]Institute of Neurobiology, National Autonomous University of Mexico, Juriquilla Querétaro, Mexico; [3]Department of Neurobiology, Duke University Medical Center, Durham, United States

**Abstract** Sucrose's sweet intensity is one attribute contributing to the overconsumption of high-energy palatable foods. However, it is not known how sucrose intensity is encoded and used to make perceptual decisions by neurons in taste-sensitive cortices. We trained rats in a sucrose intensity discrimination task and found that sucrose evoked a widespread response in neurons recorded in posterior-Insula (pIC), anterior-Insula (aIC), and Orbitofrontal cortex (OFC). Remarkably, only a few Intensity-selective neurons conveyed the most information about sucrose's intensity, indicating that for sweetness the gustatory system uses a compact and distributed code. Sucrose intensity was encoded in both firing-rates and spike-timing. The pIC, aIC, and OFC neurons tracked movement direction, with OFC neurons yielding the most robust response. aIC and OFC neurons encoded the subject's choices, whereas all three regions tracked reward omission. Overall, these multimodal areas provide a neural representation of perceived sucrose intensity, and of task-related information underlying perceptual decision-making.

DOI: https://doi.org/10.7554/eLife.41152.001

*For correspondence:
ranier@cinvestav.mx

Competing interests: The authors declare that no competing interests exist.

## Introduction

Chemical stimulation of taste receptor cells elicits signals that are transduced into neural representations of multiple attributes, such as taste quality, intensity (the strength or concentration of a stimulus), and palatability (hedonic value). These attributes form a single percept (*Accolla et al., 2007*; *Breslin, 2013*; *Lemon, 2015*) that informs the animal whether it is safe to ingest the food (*Tapper and Halpern, 1968*). Sucrose is the prototypical highly palatable tastant for sweet taste quality, and it provides a sensory cue predicting the presence of immediate energy sources. Although palatability and intensity usually change together, *Wang et al., 2018* found this is not always the case and suggested that they are two distinct representations. In rodents, palatability is measured by an increase in positive oromotor responses (e.g., licking) elicited by increasing sucrose concentrations (*Spector and Smith, 1984*). In contrast, the intensity attribute cannot be directly measured by any licking response per se, as an animal must actively report the perceived concentration of sucrose, a process necessarily involving decision-making. Historically, the neural representation of sweet taste intensity has been characterized by firing rates (spike counts) that monotonically increase with sucrose concentration along the gustatory pathway from the periphery to primary (IC) and secondary (OFC) taste cortices (*Rolls et al., 1990*; *Roussin et al., 2012*; *Scott et al., 1991*; *Thorpe et al., 1983*; *Villavicencio et al., 2018*). However, those responses were obtained in either anesthetized animals (*Barretto et al., 2015*; *Wu et al., 2015*), during passive intraoral delivery of tastants (*Maier and Katz, 2013*; *Scott et al., 1991*), or in behavioral tasks where animals do not have to make any decision other than to lick (*Rosen and Di Lorenzo, 2012*; *Stapleton et al., 2006*; *Villavicencio et al., 2018*). Thus, the neural representation of the perceived intensity of sucrose that

**eLife digest** Imagine you wake up in the morning, and you pour yourself and your loved one coffee. They like it with two sugars but you only with one. Our ability to distinguish different sweet intensities allows us to detect how much sugar is in the coffee. It also helps us to predict the amount of energy present in foods and if it is safe to ingest. We can experience the sweet quality because our tongue contains sweet taste receptor cells that are switched on by sugar. This activates neurons across our taste system in the brain.

However, we do not completely understand how these areas represent the intensity of sugar. Previous studies have only 'passively' measured different sugar concentrations, either using anesthetized animals or behavioral tasks that do not involve decision-making other than licking. But to accurately evaluate how animals perceive the intensity, active decision-making is required, such us 'reporting' the perceived concentration of sugar.

Fonseca et al. set out to answer this question by training rats in a new sweet intensity discrimination task, in which the rats had to move to the left or right to obtain water as a reward. This way, the animals could 'indicate' how sweet they perceived the sugar water to be. At the same time, recordings from the three brain areas involved in taste responses were taken (called the anterior and posterior insular cortices, and the orbitofrontal cortex) to measure how the sugar intensity is processed in the brain.

The results showed that a small group of neurons within all three areas contained more information about the sugar intensity than other neurons, suggesting the taste system uses a compact and distributed code to represent its intensity. The information about sugar intensity was contained in both the number of nerve impulses and in the precise timing with which these neurons fired.

Many drinks and high-energy foods often contain large quantities of sugar, and their overconsumption contributes to the worldwide problems of obesity and its associated diseases. Therefore, a better understanding of the neurons that code information about the intensity of sugar could be a starting point for other studies to pinpoint the connections and areas in the brain involved in our irremediable attraction for sugar.

DOI: https://doi.org/10.7554/eLife.41152.002

the animal actively reports has not presently been studied. Likewise, how this representation is transformed into perceptual decision-variables, such as choice, movement direction, and the presence or absence of reward remains to be elucidated. Here we trained rats in a sucrose intensity discrimination task and recorded electrophysiological responses in the posterior (pIC), anterior (aIC) insular cortices, and the orbitofrontal cortex (OFC), with the aim of elucidating how these cortices encode sucrose intensity and use this information to guide behavior.

These three cortical areas are multimodal and chemosensitive and are involved in disgust (pIC), tastant identification (aIC), and subjective value and reward (OFC) (*Frank et al., 2013*; *Gardner and Fontanini, 2014*; *Jezzini et al., 2013*; *Jones et al., 2006*; *Katz et al., 2001*; *Kusumoto-Yoshida et al., 2015*; *Maffei et al., 2012*; *Maier and Katz, 2013*; *Verhagen et al., 2004*). In rodents, the pIC has been shown to be involved in taste, disgust, expectancy, and aversive motivated behaviors (*Bermúdez-Rattoni, 2004*; *Chen et al., 2011*; *Fletcher et al., 2017*; *Gardner and Fontanini, 2014*; *Gutierrez et al., 2010*; *Kusumoto-Yoshida et al., 2015*; *Wang et al., 2018*). In contrast, the aIC is involved in appetitive behaviors, and besides having neurons that respond selectively to sweet taste (*Chen et al., 2011*), it also has neurons encoding reward probability and reward omission (*Jo and Jung, 2016*). Even though both pIC and aIC have roles in taste and decision-making, their contribution to sucrose intensity guided behavior remains unexplored.

It is well known that OFC is involved in reward and subjective value (*Conen and Padoa-Schioppa, 2015*; *Jo and Jung, 2016*; *Kennerley and Wallis, 2009*; *Roesch et al., 2006*), and it is a critical brain region for encoding decision-variables such as choice, movement direction, and reward omission (*Feierstein et al., 2006*; *Hirokawa et al., 2017*; *MacDonald et al., 2009*; *Nogueira et al., 2017*). However, it is not known whether OFC neurons encode decision-variables guided by sucrose

intensity. Equally unknown is how these variables are encoded along the posterior-anterior axis of the Insula.

To address these questions, we designed a novel sweet intensity discrimination task in which, to obtain a water reward, rats had to make a rightward or leftward movement based on the perceived intensity of sucrose (Cue), while single-unit recordings in the pIC (1348), or aIC (1169), or OFC (1010) were performed. We found that stimulation with sucrose evoked a widespread response in these three cortical regions, indicating a distributed detection of taste/somatosensory information. 82% of the evoked responses showed no selectivity to sucrose intensity, whereas 18% could be labeled as sucrose intensity-selective. These selective neurons conveyed the most information about sucrose's sweet intensity. Analyses of the sucrose-evoked responses revealed that, in addition to firing rates, the spike timing of neurons contains additional information about sucrose's intensity. Several differences and similarities were identified between the evoked pIC, aIC, and OFC responses. Overall, the three recorded areas similarly decoded sucrose concentration and equally tracked the outcome (reward delivery or omission). A major difference among them was that the OFC neurons carry information about behavioral choice and movement direction, earlier and with higher quality than neurons in the Insula. In summary, these data show that the perceived intensity of sucrose is fully reconstructed from the firing rate and spike timing of a small population of neurons in the pIC, aIC, and OFC.

## Results

### Behavior

Twenty-eight rats were trained in a one-drop sucrose intensity discrimination task. The trial structure is depicted in *Figure 1A*. Briefly, trained rats initiate a trial by first visiting the central port (Return). Licking at the central spout triggers the delivery of either 10 µL of 3 (Low) or 18 (High) wt% sucrose (referred to as Cue-D; Stimulus). If a rat chooses correctly (by moving to one of the two lateral ports; Response), three drops of water are delivered as a reward (Outcome). Error trials were unrewarded. Subjects achieved the learning criterion ($\geq$80% correct) in about 25 sessions (*Figure 1B*), and the implantation of an electrode array in one of the three cortical areas did not impair task performance (paired t-test before vs. after surgery; $t_{(27)} = 0.95$; p=0.35; *Figure 1B*). Once the animals learned the discrimination task, they were tested in a variant named generalization session (*Figure 1C*). In these sessions that consisted of 20% of the trials, rats were required to classify 0, 3, 4.75, 7.5, 11.75, or 18 wt% sucrose as either 'Low' or 'High' (referred as Cue-G). In these trials, no reward was delivered (Reward omission) to avoid imposing an arbitrary Low/High threshold that could bias the behavioral report of the perceived sweetness intensity. In Cue-G trials the percentage of 'High' responses increased with increasing sucrose concentration (*Figure 1D*), thus showing that the animals used sucrose intensity as a cue to solve the task (since its quality is unchanged (*Pfaffmann et al., 1979*)). Surgery did not impair perceptual judgments based on sucrose intensity (see Before vs. After surgery; *Figure 1D*).

Other behavioral measurements, related to palatability (*Perez et al., 2013*; *Spector et al., 1998*), revealed that the latency to stop licking after High Cue-D delivery (18% sucrose) was longer (0.74 ± 0.02 s) than for the Low Cue-D (3% sucrose; 0.58 ± 0.01 s; p < 0.0001). A similar trend was observed for generalization cues (i.e., Cue-G trials), in that rats exhibited a longer time to stop licking in trials where sucrose intensities were $\geq$to 4.75% relative to Low Cue-D (One-way ANOVA: $F_{(7,1360)} = 17.10$; p < 0.0001; Dunnett *post-hoc*; *Figure 1E*). Furthermore, we analyzed the relationship between licking and task performance and found that rats lick more rhythmically and similarly for both cues in sessions where their performance was better. This is reflected by a positive correlation between Low and High licking PSTHs and task performance (r = 0.17, p < 0.003; see *Figure 1—figure supplement 1*). Thus, rats did not solve the task by licking differently for both cues.

In the Return epoch, rightward movements (left to center port direction) were faster than leftward (right to center port) movements (*Figure 1F*). In contrast, during the Response epoch, leftward or rightward movements were not significantly different (*Figure 1G*), and therefore these movements were independent of the sucrose concentration. Interestingly, rats moved faster in the Response than in the Return period, perhaps a result of the water reward (compare *Figure 1F* vs. *Figure 1G*). Finally, in the Outcome epoch, rats rapidly detected when the reward was omitted (Cue-G trials and

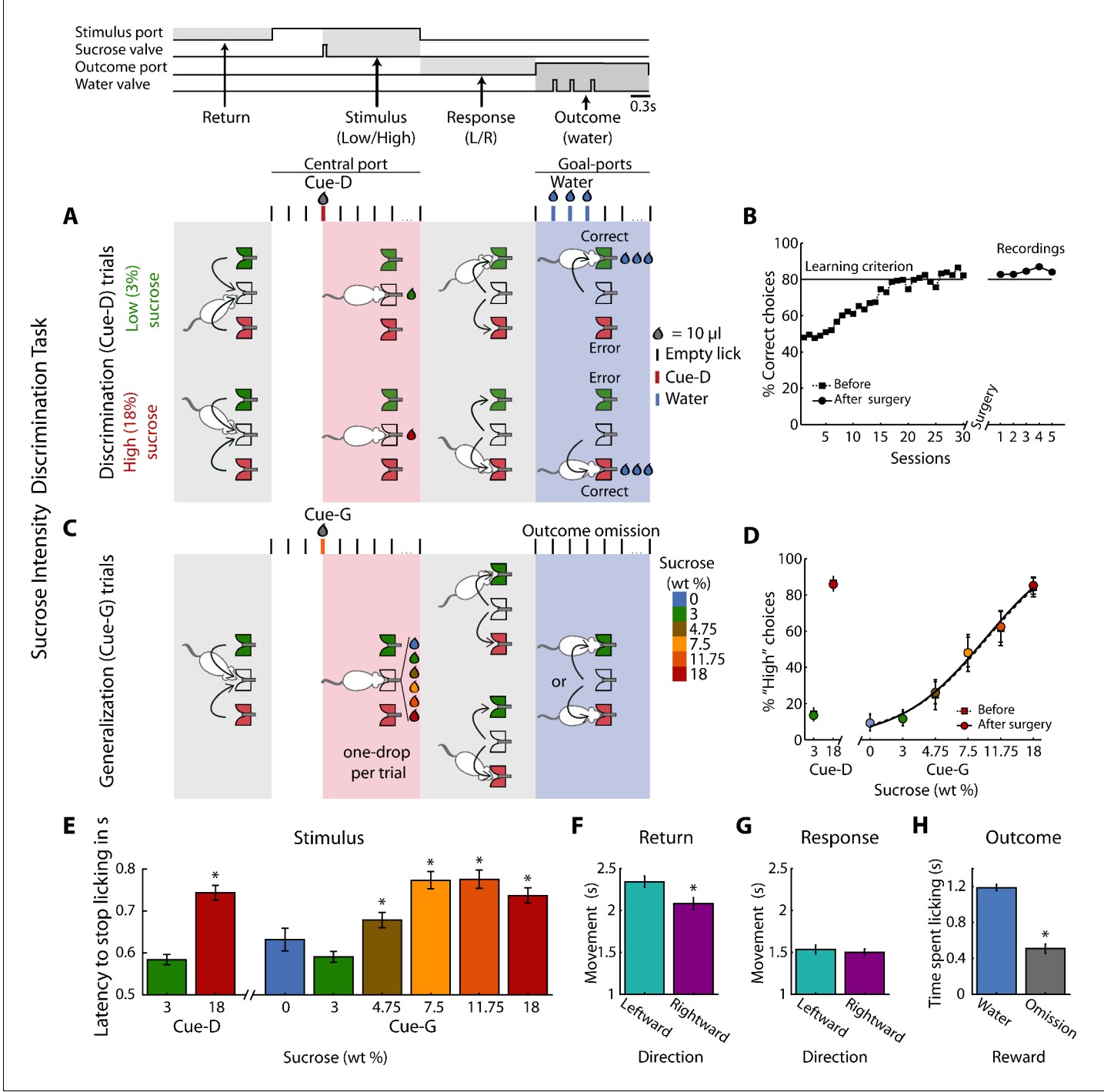

**Figure 1.** Behavioral report of sucrose's sweet intensity in a one-drop discrimination task. (**A**) Structure of a single trial. The behavioral box was equipped with three spouts each connected to a pressure-controlled solenoid valve that delivered 10 µL drops (not shown, see Materials and methods). One spout was in the central (stimulus) port and the others in the left and right lateral (choice) ports. After the first trial, in the Return epoch, animals after obtaining one of the outcomes in the lateral ports, returned to the central port to begin a new trial. In the Stimulus epoch, after two or three dry licks, the cues (Cue-D -for discrimination) were delivered, and the animals had to make a High/Low decision as to which lateral port to go (Response epoch). If they choose correctly, a water reward was delivered in the Outcome epoch. Errors were unrewarded. Thus, in this task, the perceived intensity of sucrose (i.e., concentration) served as a discriminative cue (Cue-D, see the red tick and drop). (**B**) Performance (percent correct choices) across training days before (dashed line squares) and after (circles) electrode implantation. (**C**) Interleaved in sessions, all animals were tested in a variant of the above-described intensity discrimination task -named generalization sessions. These sessions were composed of 80% discrimination trials (3% Low/ 18 wt% High) that were rewarded as indicated in 'A,' and 20% generalization trials; that is, 0, 3, 4.75, 7.5, 11.75, and 18 wt% sucrose cues (named Cue-

*Figure 1 continued on next page*

*Figure 1 continued*

G). For generalization trials, rats were required to 'classify' these sucrose concentrations as either a 'Low' or 'High,' but these trials were unrewarded. (D) The percent responses to the 'High' port during discrimination (Cue-D) and generalization (Cue-G) trials increases as the sucrose concentration increase. Note that the psychometric function was nearly identical both before (dashed line squares) and after surgery (circles). (E) Latency to stop licking after cue delivery. On average, the higher the sucrose concentration, the longer the latency to stop licking. (F–G) Movement time for making a leftward or rightward movement in the Return and Response epochs. (H) Time spent licking, in the Outcome epoch, in Cue-D trials that received water as a reward was longer then in Cue-G trials where the water reward was omitted. * Statistically significant with an alpha level of 0.05.

DOI: https://doi.org/10.7554/eLife.41152.003

The following figure supplements are available for figure 1:

**Figure supplement 1.** In sessions with rats having better performances, they licked similarly for both cues.

DOI: https://doi.org/10.7554/eLife.41152.004

**Figure supplement 2.** Extracellular recordings were obtained in either the posterior, anterior Insular cortices (pIC or aIC), or orbitofrontal cortex (OFC).

DOI: https://doi.org/10.7554/eLife.41152.005

Cue-D error trials). That is, they spent more time licking when water was delivered than when it was omitted (*Figure 1H*; see *Supplementary file 1* for statistics). In sum, by using only a 10 µL drop of sensory stimulation, rats can make accurate perceptual decisions based on the perceived concentration of sucrose.

## Electrophysiology

A total of 1348, 1169, and 1010 single-units were recorded from pIC, aIC, and OFC, respectively (see *Figure 3—figure supplement 1A*). Of these neuronal responses 480, 403, and 337, respectively were recorded in generalization sessions and the rest in discrimination sessions (with only cue-D trials). Recordings were performed unilaterally in the left hemisphere. Schematics and location of the recording sites are seen in *Figure 1—figure supplement 2*.

## Modulation profiles of Cue-D discrimination trials

The temporal activation pattern of the neural responses in pIC, aIC, and OFC was classified as a function of the evoked response (Cue-evoked or Non-evoked), modulation profile (Phasic, Tonic, or Coherent; see *Table 1*), and selectivity (either Non-selective or Intensity-selective; see *Table 2*). Most recorded neurons exhibited a statistically significant evoked response 90.6% (1221/1348), 97.4% (1139/1169), and 92.8% (937/1010) for the pIC, aIC, and OFC, respectively. The remaining neurons, named Non-evoked, were 9.4% (127/1348), 2.6% (30/1169), and 7.2% (73/1010), respectively. Cue-evoked responses were then further classified according to five characteristic modulation profiles: Phasic, Tonic-Inactive (Inact), Tonic-Active (Act), Lick-coherent Inactive (Coh-Inact), and Active (Coh-Act) (*Table 1*).

Given that rats could use a drop of sucrose to make accurate perceptual decisions based on its intensity (*Figure 1*), we explored the neural correlates of these decisions in the pIC, aIC, and OFC. *Figure 2* depicts the raster plots and corresponding peri-stimulus time histograms (PSTHs) of representative examples of Intensity-selective Cue-D evoked responses recorded in each of the three cortical regions. Examples of Non-selective Cue-D responses are shown in *Figure 2—figure supplement 1*. Action potentials are depicted as black ticks and were aligned to Cue-D delivery (time = 0 s). Trials were sorted as a function of Low (3% -green) and High (18 wt% -red). The left

**Table 1.** Cue-Evoked and Non-Evoked neurons.

| Brain region | Cue-Evoked responses | | | | | Non-Evoked | |
| --- | --- | --- | --- | --- | --- | --- | --- |
| | Phasic | Inactive | Active | Coh-Inac | Coh-Act | Non-Mod | Coh-NonEvo |
| pIC (n=1348) | 75 (5.6) | 217 (16.1) | 193 (14.3) | 414 (30.7) | 322 (23.9) | 53 (3.9) | 74 (5.5) |
| aIC (n=1169) | 67 (5.7) | 202 (17.3) | 192 (16.4) | 317 (27.2) | 361 (30.9)* | 7 (0.6)* | 23 (2)* |
| OFC (n=1010) | 27 (2.7)*# | 386 (38.2)*# | 265 (26.2)*# | 169 (16.7)*# | 90 (8.9)*# | 62 (6.1)*# | 11 (1.1)* |

Number of Cue-D responsive neurons (%). Data in bold indicate statistically different against pIC (*) or aIC (#), detected by a chi-squared test. Alpha level set at 0.05.

DOI: https://doi.org/10.7554/eLife.41152.006

**Table 2.** Cue-Evoked responses: Non-selective and Intensity-selective neurons.

| Brain region | Cue-Evoked responses | | | | | | | | | | | |
| --- | --- | --- | --- | --- | --- | --- | --- | --- | --- | --- | --- | --- |
| | Phasic | | Inactive | | Active | | Coh-Inac | | Coh-Act | | Total | |
| | Non-Sel | Int-Sel | Non-Sel | Int-Sel | Non-Sel | Int-Sel | Non-Sel | Int-Sel | Non-Sel | Int-Sel | Non-Sel | Int-Sel |
| pIC (n=1348) | 55 (4.1) | 20 (1.5) | 198 (14.7) | 19 (1.4) | 171 (12.7) | 22 (1.6) | 325 (24) | 89 (6.6) | 272 (20.2) | 50 (3.7) | 1021 (75.7) | 200 (14.8) |
| aIC (n=1169) | 55 (4.7) | 12 (1) | 183 (15.7) | 19 (1.6) | 155 (13.3) | **37 (3.2)\*** | 254 (21.6) | 63 (5.4) | 283 (24.2) | **78 (6.7)\*** | 930 (79.6) | 209 (17.9) |
| OFC (n=1010) | **21 (2.1)\*#** | **6 (0.6)\*** | **336 (33.3)\*#** | **50 (5)\*#** | **204 (20.2)\*#** | **61 (6)#** | **124 (12.3)\*#** | **45 (4.5)\*** | **65 (6.4)\*#** | **25 (2.5)#** | 750 (74.2) | **187 (18.5)\*** |

Number of Non-selective and Intensity-selective neurons (%). Data in bold indicate statistically different against pIC(*) or aIC (#) detected by a chi-squared test. Alpha level set at 0.05.

DOI: https://doi.org/10.7554/eLife.41152.007

column shows three different neurons that exhibited selective phasic responses to 3 wt% sucrose in the pIC, aIC, and OFC, respectively. Examples of the Tonic-Inactive (second column) revealed a selective inhibition for 3 wt% sucrose (Low-preferred). After Cue-D delivery, the Tonic-Active neurons exhibited a sustained increase in firing rate (third column; the upper and lower panels depict a Low-preferred neuron, whereas the middle panel a High-preferred response). The last two columns on the right-hand side display examples of neurons that fired synchronously with licking (named Lick-coherent) and, after Cue-D delivery, exhibited either a decrease (Coh-Inact) or an increase (Coh-Act) in their firing rate.

Intensity-selective neurons were recorded in all three cortical regions and for all five classes of evoked responses, although with different proportions (see *Figure 3—figure supplement 1A* and *Table 2*; see *Supplementary file 2* for statistics). In general, pIC and aIC Intensity-selective neurons exhibited more similar responses between them than those found in the OFC (see *Table 2*). The only exception was that the aIC contained more Intensity-selective neurons with Tonic-Active and Coh-Act responses than the pIC. In contrast, the OFC had more Intensity-selective neurons exhibiting Tonic-Inactive and Active responses (*Table 2*). Overall, the percentage of Intensity-selective neurons were 14.8%, 17.9%, and 18.5%, in the pIC, aIC, and OFC, respectively (see *Table 2*; Total, Inten-Sel). These data show that Intensity-selective neurons are found along the posterior-anterior taste neuroaxis.

To determine, in fine-grain detail, the differences in licking and its impact upon neuronal responses, in *Figure 2*, we also depicted the corresponding PSTHs of licking behavior and the times where the lick rate was significantly different between Low and High cues (see dashed lines). We found that 45.1% of all Intensity-selective neurons have a 'best-window' (interval with maximal discrimination between concentrations) with no differences in licking (see *Figure 2* grey-line above the PSTHs). The remaining 54.9% of neurons have a lick rate difference inside the best-window, but most frequently they only covered a small fraction of the window (*Figure 2—figure supplement 2*). Specifically, the overlap of the lick rate differences covered 31.4% of the entire best-window (*Figure 2—figure supplement 2*). Thus, we conclude that is unlikely that most sucrose intensity representation can be attributed to differences in licking behavior.

*Figure 3A* shows the color-coded population PSTH of the responses of all Intensity-selective neurons in each brain region, sorted as a function of the modulation profile and preferred concentration. What is clear in the figure is that diverse temporal patterns are evoked in response to the delivery of the Cue-D (time = 0 s). The evoked responses can be transient, sustained, or oscillatory, with either increasing or decreasing firing rates.

In the Stimulus epoch, the population responses revealed that the pIC and aIC were more excited, whereas the OFC was inhibited (*Figure 3—figure supplement 1B*), suggesting an opposite interaction between the Insula and OFC during licking behavior. In agreement with the idea that in a default brain-network state, these two brain regions function out of phase (*Gutierrez-Barragan et al., 2018*). In line with previous studies (*de Araujo et al., 2006*; *Gutierrez et al., 2010*), we found among these taste cortices that the pIC (60.3%) and aIC (59.5%) had a higher proportion of (either increasing or decreasing) lick-induced oscillatory responses than the OFC (27.6%). Likewise, we found that the coherence values of the OFC (0.24 ± 0.005) were significantly lower relative to pIC (0.26 ± 0.003) and aIC (0.26 ± 0.003) ($F_{(2, 1672)}=3.77$; p = 0.02) (*Figure 3—figure supplement 3A*). Therefore, the pIC and aIC had not only a higher proportion of Lick-coherent neurons than

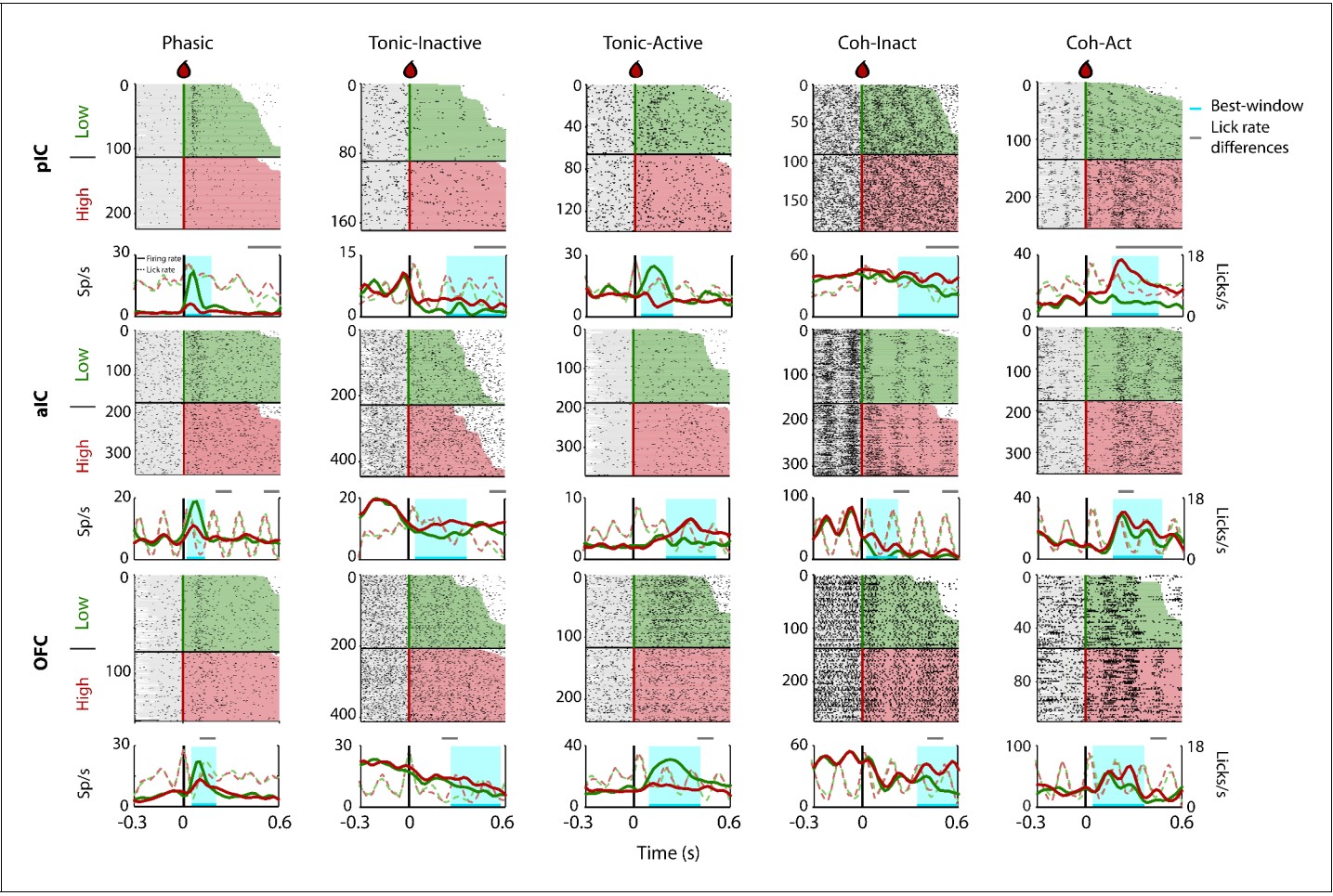

**Figure 2.** Representative Intensity-selective Cue-evoked responses in the rat pIC, aIC, and OFC. Representative raster plots and PSTHs (in spikes/s solid lines) of sucrose Intensity-selective neurons belonging to each of the five classes of evoked responses in the pIC (upper), aIC (middle), and OFC (lower) rows: Phasic, Tonic-Inactive, Tonic-Active, Coh-Inactive, and Coh-Active. Coh indicates they are coherent with licking. These exemplar neuronal responses discriminated between 3 and 18 wt% sucrose (Intensity-selective neurons). Most of the Cue-evoked responses were Non-selective to sucrose intensity, and individual examples are presented in **Figure 2—figure supplement 1**. Action potentials are depicted as black ticks around −0.3 to 0.6 s, from Cue-D delivery (time = 0 s). Only correct trials were included in these plots. The horizontal black line separates the sorted trials according to Cue-D delivery. The licks after 3 and 18 wt% sucrose are indicated by green- and red-shaded area, respectively. The times that animals were licking at the central spout before cue delivery are shown in the shaded gray areas. Also shown are the PSTHs for licking (Licks/s; at right axis) either for Low (green-dashed) or High sucrose (red-dashed line). The rectangle in cyan highlights the best-window where the responses to 3 and 18 wt% sucrose are statistically distinct as determined by a Wilcoxon rank-sum test. The gray horizontal line on top indicates the times where the lick rates were significantly different.

DOI: https://doi.org/10.7554/eLife.41152.008

The following figure supplements are available for figure 2:

**Figure supplement 1.** Representative Cue-evoked non-selective responses in the pIC, aIC, and the OFC.
DOI: https://doi.org/10.7554/eLife.41152.009

**Figure supplement 2.** Times in % where the lick rate differences overlapped the best-window.
DOI: https://doi.org/10.7554/eLife.41152.010

OFC, but also IC neurons were better entrained with rhythmic licking. More importantly, we also uncovered, for the first time, that the level of coherence was significantly higher in the Stimulus-epoch in comparison with the pre-Stimulus and the Outcome epochs (all $p$'s < 0.0001), suggesting that lick-spike coherence reflects more than oromotor responses, perhaps it prepares taste cortices to receive sensory inputs.

For the neuronal populations of each brain region, a linear decoder was used to estimate the accuracy for discriminating Low and High sucrose trials (**Meyers, 2013**; see Materials and methods).

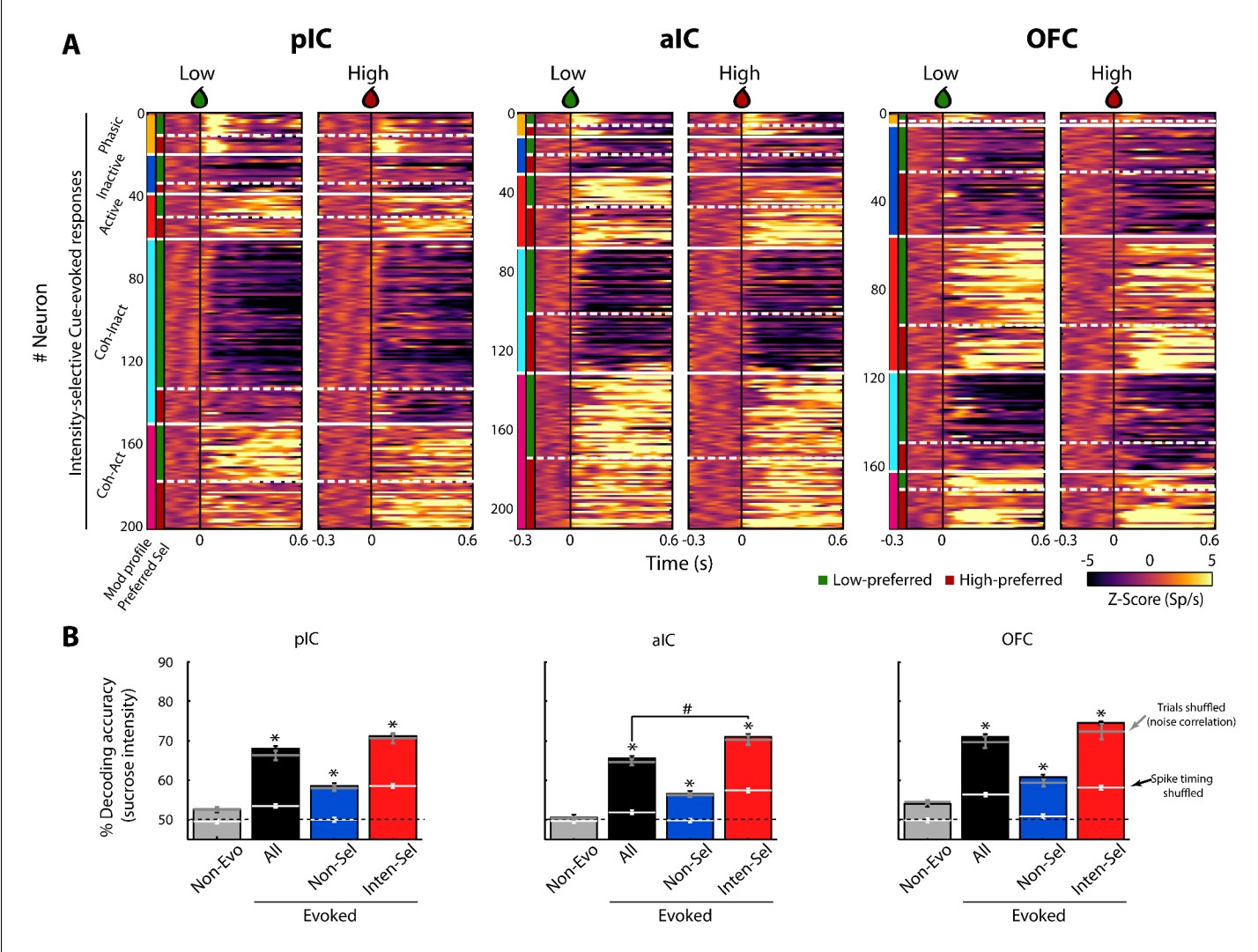

**Figure 3.** A subpopulation of Intensity-selective neurons decodes sucrose concentrations (3 vs. 18 wt%) better than other neuron classes. (**A**) The color-coded PSTHs of the five Cue-evoked responses in pIC (left panel), aIC (middle panel), and OFC (right panel) sorted by modulation profile and Intensity-selectivity (Low/High). Response types were first sorted from top to bottom as follows: Phasic (orange vertical line at the left edge), Inactive (dark blue), Active (red), Lick-coherent Inactive (cyan), and Lick-coherent Active (magenta). The white horizontal dashed lines separate each modulation profile as a function of Low and High selectivity (see green and red vertical lines at the left edge). Each row represents the firing rate normalized in Z-score of a neuronal response aligned to 3 (Low, left panel) and 18 wt% (High, right panel) sucrose delivery (time = 0, black vertical line). (**B**) Percent decoding accuracy of sucrose intensity achieved by the neurons recorded in pIC, aIC, and OFC. Each colored bar represents a different group of neurons: Non-evoked (gray), All (black), Non-selective (blue), and Intensity-selective (red). A black dashed line indicates the 50% chance level, and the upper dashed line the behavioral performance. * Indicates significant differences against the Non-evoked population, while # indicates significant differences against All group. Only correct Cue-D trials were included for analysis. The white horizontal line in each bar indicates the percent decoding achieved by each population when spike timing information was removed (i.e., shuffled spikes but maintaining same firing rates). The gray horizontal lines depict the contribution of noise correlations for population decoding.

DOI: https://doi.org/10.7554/eLife.41152.011

The following figure supplements are available for figure 3:

**Figure supplement 1.** The population temporal activation pattern of all neurons recorded in the posterior IC (n = 1348), anterior IC (n = 1169), and the OFC (n = 1010), elicited in rats by a single10 μL drop of 3 and 18 wt% sucrose.
DOI: https://doi.org/10.7554/eLife.41152.012

**Figure supplement 2.** Noise correlation is removed after shuffling the order of trials.
DOI: https://doi.org/10.7554/eLife.41152.013

**Figure supplement 3.** The coherence between licks and spikes is larger after Cue-D delivery in the Stimulus epoch than in pre-Stimulus and Outcome.
DOI: https://doi.org/10.7554/eLife.41152.014

As seen in *Figure 3B*, the contribution of the Non-evoked responses (grey bars) was found to be at chance level (50%), indicating that they contained little, if any, information about sucrose intensity. In contrast, all Cue-D evoked neuronal responses (All- black bars) significantly decoded sucrose concentrations above chance level (*Figure 3B*). Importantly, we found that the small population of Intensity-selective neurons (red bars) contained more information than the larger Non-selective population (blue bars).

Interestingly, the Non-selective group also decoded sucrose intensity significantly above chance level. One possibility is that they have subtle differences in firing rates that are not consistent enough across trials to produce a significant effect at single neuronal level. However, at the population level there is sufficient information about sucrose intensity. Alternatively and despite their similar firing rates (spike counts) evoked by Low and High Cue-D, these neurons could use spike timing to discriminate sucrose concentrations (*Gutierrez and Simon, 2013*). To test this hypothesis, the spikes of all Non-selective neurons were shuffled without changing their average firing rates. When the spike timing information was eliminated from these neuronal responses, their ability to decode among the sucrose's intensities dropped to chance level (*Figure 3B*; see the horizontal white lines across the blue bars). Thus, the additional information in the Non-selective population was likely conveyed by precise spike timing patterns of activity.

The decoding algorithm also revealed that the Intensity-selective neurons in the three cortical regions decoded sucrose intensity better than Non-selective neurons (*Figure 3B*; red bars). It is unlikely that these results were due to the differences in the population size since Intensity-selective neurons were always fewer in number than the Non-selective and All Cue-evoked neurons. Thus, the Intensity-selective population (i.e., less than 18% of neurons) contained more information about sucrose intensity than the entire population. These data suggest the existence of a neuronal representation of sucrose concentration across these three gustatory cortical regions. That is, each taste cortex seems to contain a copy of this information. Finally, we note that by removing the spike timing information contained in the Intensity-selective neurons, their percent decoding dropped to nearly chance level, indicating that the neural representation of sucrose intensity is also conveyed in the spike timing of neurons.

It has been reported that spike counts in a pair of simultaneously recorded neurons, elicited by a stimulus, can covary across the session, a phenomenon denominated as noise-correlation and these correlations are thought to covary with attentional, behavioral, and overall brain-state of the network (*Averbeck et al., 2006*). Although the function of noise-correlations is not completely understood, it is well known that they could affect (either increase or decrease) population decoding (*Averbeck et al., 2006*; *Averbeck and Lee, 2006*; *Carnevale et al., 2013*; *Cohen and Kohn, 2011*; *Zohary et al., 1994*). For this reason, we also determined the impact of removing the noise-correlations on the decoding accuracy of sucrose intensity. We found that pIC ($0.21 \pm 0.005$) had a significantly higher noise-correlations in comparison to aIC ($0.19 \pm 0.006$) and OFC ($0.19 \pm 0.007$) ($F_{(2,386)} = 7.85$; $p = 0.0005$; *Figure 3—figure supplement 2A*). Nevertheless, removing noise-correlations by shuffling trials (*Figure 3—figure supplement 2B*) did not significantly affect decoding accuracy in any population or recorded brain region (*Figure 3B* see the grey horizontal lines). Therefore, at least in these experiments, noise-correlations do not have a significant effect over decoding of sucrose intensity.

We next determined which class of Cue-evoked responses contained sufficient information to decode sucrose's intensity. To achieve this, we ran the neural classifier using a single neuronal population. In all three regions, the Coherent-Inactive and Coherent-Active had better percentage

**Table 3.** Percent decoding accuracy of sucrose's intensity, when only one a single population, at a time, was included in the analysis.

| Brain region | Non-Evo | All | Phasic | Inactive | Active | Coh-Inac | Coh-Act | All-Tonic | All-Coh |
|---|---|---|---|---|---|---|---|---|---|
| pIC | $52.7 \pm 0.6$ | $\mathbf{67.6 \pm 1.3}$ | $\mathbf{60.2 \pm 1.7}$ | $53.1 \pm 0.5$ | $53.5 \pm 0.6$ | $\mathbf{62.0 \pm 1.2}$ | $\mathbf{61.6 \pm 0.8}$ | $54.3 \pm 0.6$ | $\mathbf{66.6 \pm 1.2}$ |
| aIC | $50.4 \pm 0.3$ | $\mathbf{65.7 \pm 0.9}$ | $\mathbf{54.7 \pm 1.1}$ | $52.1 \pm 0.4$ | $55.4 \pm 0.4$ | $\mathbf{60.0 \pm 0.7}$ | $\mathbf{62.1 \pm 0.8}$ | $55.1 \pm 0.5$ | $\mathbf{65.6 \pm 0.9}$ |
| OFC | $54.8 \pm 0.5$ | $\mathbf{71.0 \pm 1.6}$ | $56.6 \pm 1.4$ | $55.3 \pm 0.6$ | $\mathbf{61.1 \pm 1.1}$ | $\mathbf{62.7 \pm 1.1}$ | $\mathbf{65.9 \pm 1.5}$ | $\mathbf{60.8 \pm 1.0}$ | $\mathbf{69.3 \pm 1.6}$ |

Percent decoding accuracy (mean ± sem). Data in bold indicate statistically different relative to Non-Evoked neurons by using a one-way ANOVA and a Dunnett´s post hoc. Alpha level set at 0.05.
DOI: https://doi.org/10.7554/eLife.41152.015

decoding accuracy than the Non-evoked control group (*Table 3*). Moreover, in the pIC, aIC, and OFC combining All-Coherent neurons (All-Coh) achieved the best sucrose decoding nearly matching that of the entire population (All). Thus, Lick-coherent populations contained sufficient information in their responses to decode sucrose intensity.

Finally, to further determine which population contained information necessary to decode sucrose intensity, we performed a dropped population analysis (*Gutierrez et al., 2006*). In this analysis, only one population at a time was removed, and its decoding accuracy was compared against the decoding achieved by All the Cue-evoked populations combined (*Table 4*; referred as 'All'). In the three cortical regions, the percent decoding accuracy was significantly reduced only when the two Lick-coherent groups were dropped from the entire population (compare All-Coh vs. 'All;' *Table 4*). In sum, both analyses suggest that the neural responses of the Lick-coherent neurons were both sufficient and necessary to decode sucrose intensity information.

### Coding profile

Having described the modulation profile evoked by two sucrose concentrations, we next characterized whether neurons in the three recorded cortices encode sucrose's concentration-dependent information and decision-variables. We first describe neuronal responses that encode information about sucrose's concentrations. This is followed by neuronal responses that correlate with animal choices ('Choice neurons'), Direction (neurons with responses selective to either leftward or rightward movements), and finally, neurons that track the Outcome (responses that indicate the presence or absence of reward). We also discuss the overlapping among these populations.

### Concentration-dependent sucrose responses (Cue-G trials)

To determine if there was a neuronal subpopulation that tracked sucrose concentrations among the Intensity-selective (Cue-D) neurons, we evaluated neural responses during generalization trials (Cue-G: 3, 4.75, 7.5, 11.75, or 18% sucrose). In these sessions, we recorded a subpopulation of 480, 403, and 337 neurons from the pIC, aIC, and OFC, respectively. Similar to the Cue-D sessions, in the generalization sessions, we found that $94.1 \pm 1.3\%$ could be classified as Cue-evoked neurons and that from these, $83.3 \pm 1.5\%$ were Non-selective and $16.7 \pm 1.5\%$ were Intensity-selective. From the Intensity-selective population, the proportion that tracked the sucrose concentration (either positively or negatively) was 28.8% (19/66), 36.1% (26/72), and 32.3% (20/62) in the pIC, aIC, and OFC, respectively. *Figure 4A* shows raster plots and PSTHs of three representative neurons recorded in the pIC, aIC, and OFC whose responses increased with sucrose's concentrations. That is, during the 'best window' (cyan-shaded rectangle), these neurons responded with increasing activity to increasing sucrose concentrations (see Insets). We also identified neurons with an activity that negatively correlated with sucrose concentrations (*Figure 4B*). The population activity of all neurons with increasing (red) or decreasing (blue) responses was similar across the three cortical regions (*Figure 4C*). Thus, all three of these cortical areas have neurons that track the sucrose concentration.

### Neurons involved in choices

Having demonstrated how sensory information about the sucrose concentration is encoded, we next identified neurons whose activity correlated with the animal's behavioral choices. For this, in the Stimulus and Response epochs, we calculated the correlation between the neuronal activity and the perceptual intensity choices made by the animals on a trial-by-trial basis. To quantify the extent to

**Table 4.** One population dropped analysis: Decoding sucrose intensity excluding only one population from All neurons.

| Brain region | All | Phasic | Inactive | Active | Coh-Inac | Coh-Act | All-Tonic | All-Coh |
|---|---|---|---|---|---|---|---|---|
| pIC | 67.8 ± 1.3 | **58.5 ± 1.7** | 68.1 ± 1.3 | 68.5 ± 1.3 | **63.4 ± 0.9** | 64.0 ± 1.2 | 68.5 ± 1.2 | **58.0 ± 0.7** |
| aIC | 65.9 ± 0.8 | 63.4 ± 2.0 | 66.3 ± 0.9 | 65.1 ± 0.9 | **62.7 ± 0.7** | **61.2 ± 0.7** | 66.1 ± 0.9 | **55.8 ± 0.5** |
| OFC | 71.2 ± 1.7 | **63.1 ± 3.0** | 72.6 ± 1.7 | 68.8 ± 1.5 | 67.6 ± 1.5 | 66.9 ± 1.3 | 70.5 ± 1.6 | **62.1 ± 1.1** |

Percent decoding accuracy (mean ± sem). Data in bold indicate statistically different in comparison with the All subpopulation by using a one-way ANOVA and a Dunnett's post hoc.

Note that values in the 'All' group are not identical to those in *Table 3* due to random sampling in the population decoder.

DOI: https://doi.org/10.7554/eLife.41152.016

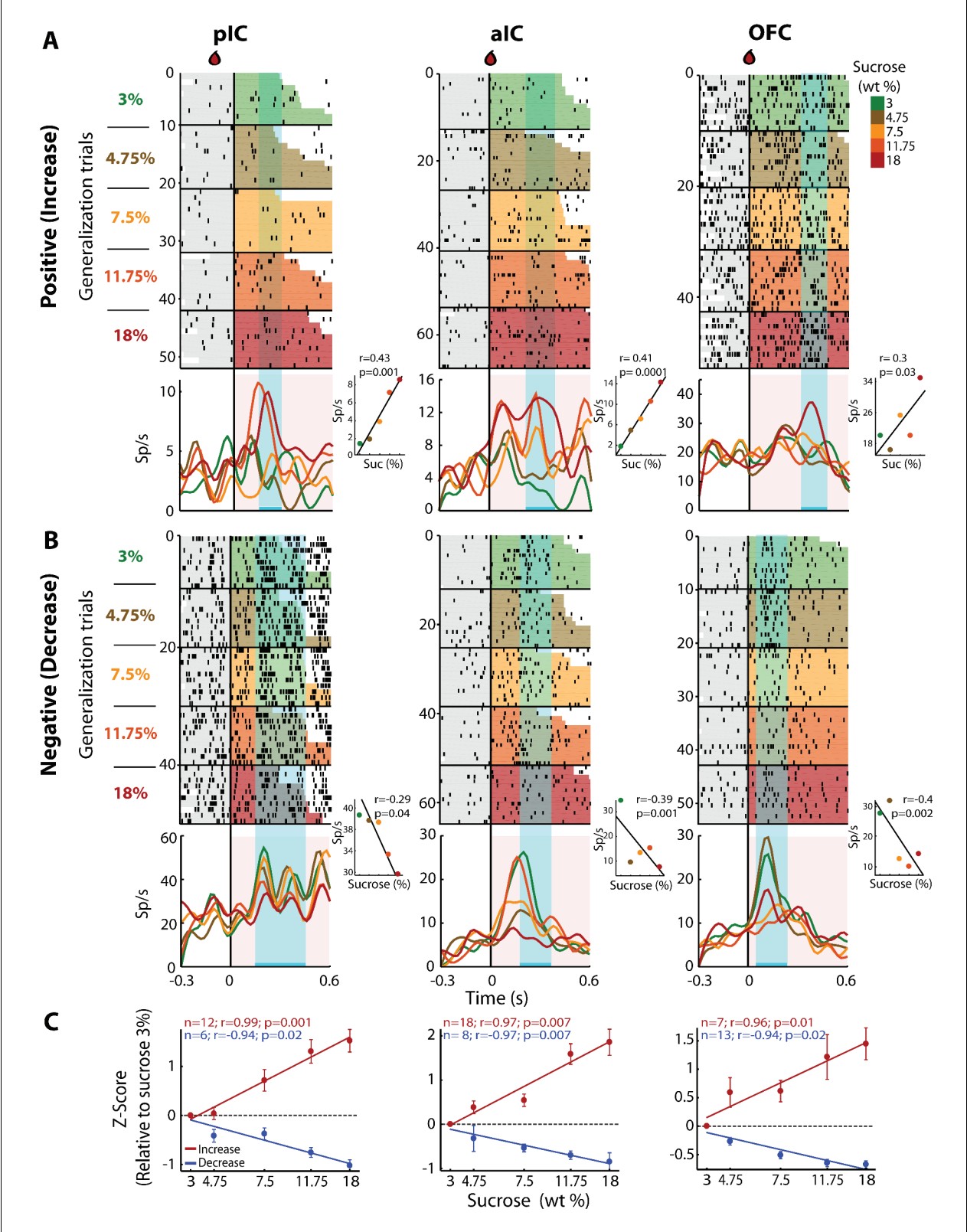

**Figure 4.** pIC, aIC, and OFC neurons track sucrose concentrations with either increasing or decreasing firing rates. (**A**) Raster plots and PSTHs of three representative Sensory neurons, with a positive correlation, recorded in pIC (left), aIC (middle), and OFC (right panel). Responses were aligned to Cue-G delivery (i.e., 3, 4.75, 7.5, 11.75, 18 wt%). The colormap on the right indicates the five sucrose concentrations delivered in the generalization trials. Each row represents a single trial and trials were sorted according to the sucrose concentration. The cyan-shaded rectangles display the 'best

*Figure 4 continued on next page*

*Figure 4 continued*
window,' in which the firing rates best correlated with sucrose concentrations (see Materials and methods for additional details). Same conventions as in *Figure 2*. The Insets displayed the firing rates in the 'best window' where responses had the best Pearson correlation coefficient against sucrose concentrations. (B) Representative examples of chemosensory neurons with negative correlation, recorded in pIC, aIC, and OFC. Same conventions as in A. (C) Normalized activity (relative to the 3 wt% trials) of all Sensory neurons that correlated either positively (red) or negatively (blue) with increasing sucrose's concentrations. Only generalization trials (Cue-G) were included in the analysis.
DOI: https://doi.org/10.7554/eLife.41152.017

which the neuronal responses could be underlying the behavioral decisions we compared psychometric and neurometric generalization curves (see Materials and methods and de Lafuente and Romo, 2005). Initially, we aligned the responses to the onset of the Stimulus epoch, but no significant neuronal responses were detected in this epoch (data not shown). In contrast, Choice-related responses were found in the Response epoch in the aIC and the OFC. Since in the pIC only two Choice neurons were detected (*Figure 5—figure supplement 1B*), no conclusions were drawn for this cortical area. The left panel of *Figure 5A* shows the PSTHs of a 'Low-preferred' aIC choice neuron (left panel) whose activity decreased with increasing sucrose concentrations. The right panel shows a 'High-preferred' OFC neuron that exhibits higher firing rates for trials $\geq$ 4.75% sucrose and that fired less for $\leq$3% sucrose (*Figure 5A*, right panel). The cyan-shaded rectangle in the PSTHs depicts the window where neural responses best-predicted animal's choices. It is seen that especially in OFC, the resulting neurometric function followed the psychometric function (*Figure 5A*, insets). The averaged neurometric function of all 8 aIC (of 403; 2.0%) and 18 OFC (of 337; 5.3%) Choice-related neurons that covaried significantly with the behavioral psychometric function are shown in the left and right panel of *Figure 5B*, respectively. However, only in the OFC were the confidence intervals of the slopes overlapped, indicating that neuronal responses in this area better followed the behavioral choices, in comparison to the aIC. To determine the temporal dynamics of choice selectivity (Low vs. High), we plot a ROC index across the Response epoch (*Figure 5C*). We observed that aIC neurons encoded the choice only when the rat is responding (time >0 s), while OFC Choice neurons discriminated between sucrose concentrations before subjects started to communicate their choice. That is, OFC neurons encoded the subject's choice while the animals were still licking in the central port (time <0 s). In sum, OFC neurons carry information about sucrose intensity judgment earlier than aIC neurons.

Instead of using sucrose intensity, we note that the animals could be using palatability to generate their behavioral responses. To investigate this possibility, we used the sucrose-evoked lick rate to construct a palatability generalization function that we then compared with the psychometric generalization function. The results show that licking responses could be used to predict behavioral responses in only 1 out of 171 sessions. Moreover, no Choice-related neurons were recorded from this session (data not shown). Therefore, we consider it is unlikely that rats guided their choices based on oromotor sucrose -evoked palatability responses but rather favor the idea that rats make decisions based on sucrose's intensity.

## Preferred direction neurons

Previous studies have demonstrated the existence of neurons in the OFC that encoded information about movement direction (*Feierstein et al., 2006*; *MacDonald et al., 2009*; *Roesch et al., 2006*). To both confirm and extend those studies we determined if there was a similar movement-direction coding in the pIC, aIC, and OFC. This was accomplished by employing a Receiver Operating Characteristic (ROC) curve (*Green and Swets, 1966*) which determined how distinguishable were the firing rate distributions of two events (i.e., leftward vs. rightward movement). The area under the ROC curve was scaled from $-1$ to $+1$, providing a Preference Index ($P_{index}$), where $-1$ means a complete preference for leftward direction, $+1$ a complete selectivity toward the rightward direction, and 0 indicates no preference. Then, using the firing rates, we computed $P_{index's}$ for the Return and Response epochs.

In the Response epoch, rats moved from the center port to a lateral port (left/right) whereas in the Return epoch go from a lateral port to central port (left/right; see schematics *Figures 1A* and *6A*). Thus, both epochs shared a similar, leftward or rightward, movement direction. We reasoned that Direction-selective neurons should fire for movements sharing a direction, but that may occur at

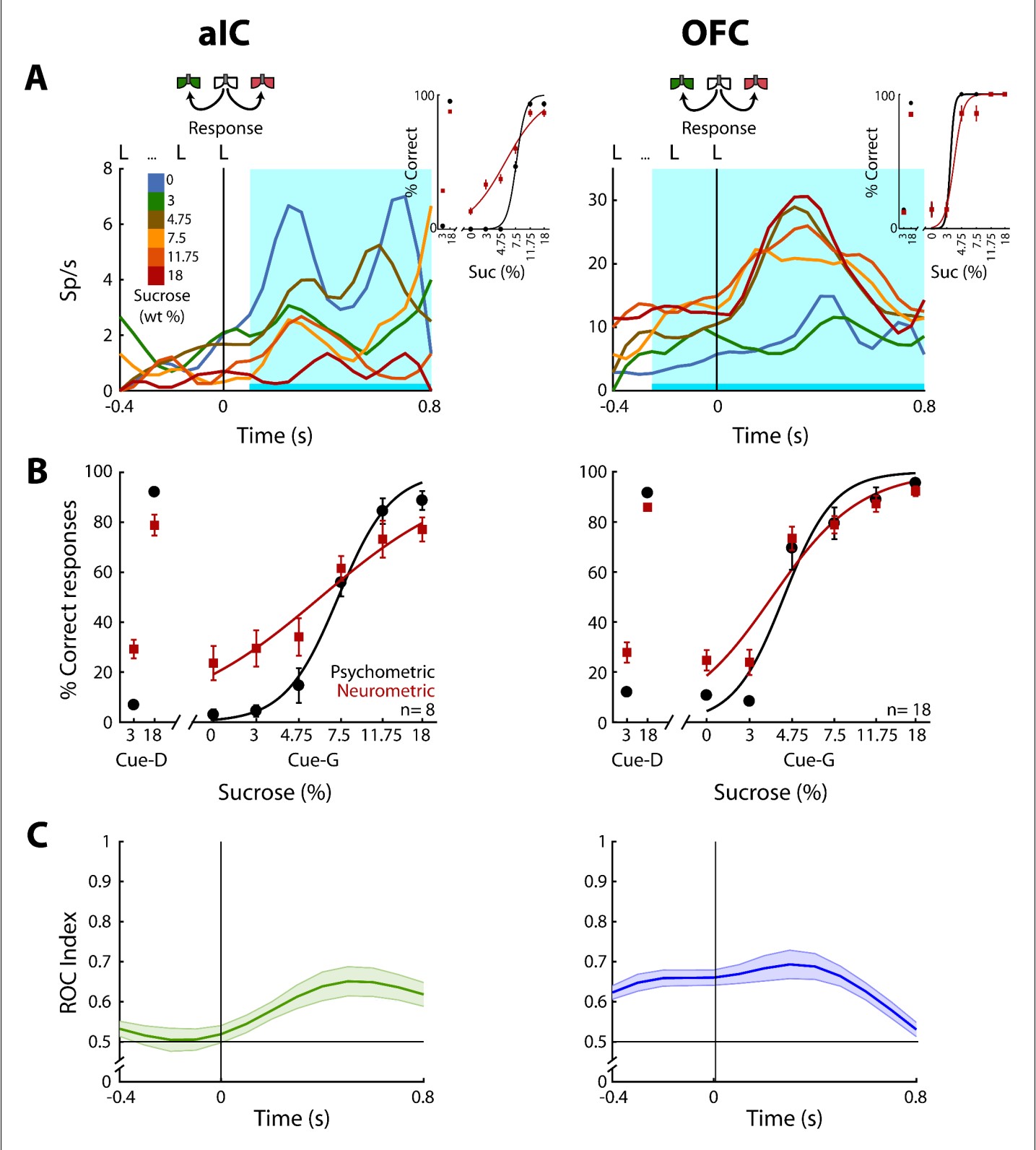

**Figure 5.** OFC Choice neurons carried information about the subjects' decisions earlier than in the aIC. (**A**) Examples of an aIC (left panel) and an OFC (right panel) neuronal response aligned to the last lick given at the central spout. Firing rates covaried with subject's Low or High choices. After the last lick in the central spout, subjects must walk to one of the lateral spouts. The colormap indicates the sucrose concentration. The PSTHs show the firing rates before and after rats had initiated the response movement. The cyan-shaded rectangle indicates the 'best window' where neural activity tracked

*Figure 5 continued on next page*

*Figure 5 continued*

subject's choices, see Inset for psychometric (black) and neurometric (red) functions of each neuron. (B) Mean percentage of correct behavioral responses (psychometric curve, black sigmoid) and neuronal responses (neurometric curve, red sigmoid) of 8 aIC and 18 OFC neurons that tracked choice along the Response epoch. Responses to discrimination (Cue-D) and generalization (Cue-G) trials are depicted on the left and right side of the sigmoid, respectively. Data are expressed as mean ± sem. (C) ROC index across the Response epoch for the aIC (left) and the OFC (right panel) neurons.

DOI: https://doi.org/10.7554/eLife.41152.018

The following figure supplement is available for figure 5:

**Figure supplement 1.** OFC Choice neurons display a better accuracy (% correct responses) than pIC and aIC neurons.

DOI: https://doi.org/10.7554/eLife.41152.019

different spatial locations. In the three cortices studied, we identified neurons that exhibited direction selectivity. *Figure 6A* shows three neuronal responses that exhibited either a Leftward-preferred selectivity (upper and lower panel) or a Rightward-preferred selectivity (middle panel). That is, the Leftward-preferred responses increased for leftward movements and did not respond to rightward movements (*Figure 6A*, cyan PSTHs). The middle panel shows a neuronal response from the aIC that fired better for a Rightward movement. The three panels in *Figure 6B* show, for all Direction-selective neurons in the three areas, the scatter plot of the Return's $P_{index}$ relative to the Response's $P_{index}$. The black arrows indicate the $P_{indexes}$ for the three representative neurons shown in *Figure 6A*. Note that $P_{index}$ values closer to the diagonal denote similar direction selectivity for the Return and Response epochs.

We also found that Direction-selective neurons displayed similar responses for Correct and Error trials (see raster plots), supporting the notion that movement direction was the primary feature modulating their firing rates. We note that the OFC had the best representation of direction selectivity tuning (Response's $P_{index}$; One-way ANOVA: $F_{(2, 462)} = 6.1$; $p < 0.0001$). A Bonferroni *post hoc* confirmed that OFC had a better representation of direction selectivity in comparison to pIC and aIC ($p < 0.05$ and $p < 0.01$, respectively). Another more complete example of direction selectivity can be seen in its population activity in both task epochs (*Figure 6C*; also see the magnitude of Z-scores), with OFC yielding the greatest differences. To this point, a higher proportion of Direction-selective neurons was found in OFC (19.1%) in comparison to pIC and aIC (10.8%, $\chi^2 = 23.85$, $p < 0.0001\%$ and 10.8%, $\chi^2 = 22.32$, $p < 0.0001$, respectively; *Figure 6B*). Note that most of these neurons were Right-Selective neurons (Insets *Figure 6B*) perhaps because we recorded unilaterally in the left hemisphere. Overall, these data reveal that more OFC neurons tracked movement direction in comparison to both areas of the IC.

## Outcome responsive neurons

Once the subjects are in the lateral goal-port, they would or would not receive water according to their choice (Correct or Error) and trial type (discrimination (Cue-D; water) or generalization (Cue-G; no water)). Recall that in Cue-D trials, reward delivery depends upon task performance whereas, in Cue-G trials, no reward was delivered regardless of choice. Thus, for Cue-G trials of 3 and 18 wt% sucrose, rats could not predict if the reward would be delivered or omitted. Therefore, by analyzing all rewarded vs. unrewarded trials (regardless of choice), we could disambiguate whether neurons tracked the outcome. In this regard, we identified a subpopulation of neurons that selectively fired for reward omission vs. reward delivery. *Figure 7A* displays the raster plots and PSTHs of three representative neurons. The pIC and OFC neurons did not respond to reward omission (RWO-named the Inactive population), but they fired to rewarded trials (RW- see dashed PSTHs). In contrast, the aIC neuron fired after reward omission (named Active population), while no responses were observed during reward delivery (*Figure 7A*, middle panel). Note that the pIC (57.1% vs. 17.9%) and the aIC (45.7% vs. 28.3%) had a higher proportion of neurons with Inactive than Active responses after reward omission ($\chi^2 = 72.88$, p<0.0001 and $\chi^2 = 12.05$, p = 0.0005; respectively); while the OFC the proportion was similar (36.5% vs. 42.4%; $\chi^2 = 1.1$, n.s.), suggesting that pIC and aIC exhibited a bias toward having more neurons with Inactive responses after reward omission relative to OFC neurons. The population responses of both Inactive and Active Reward Omission neurons are seen in *Figure 7B*.

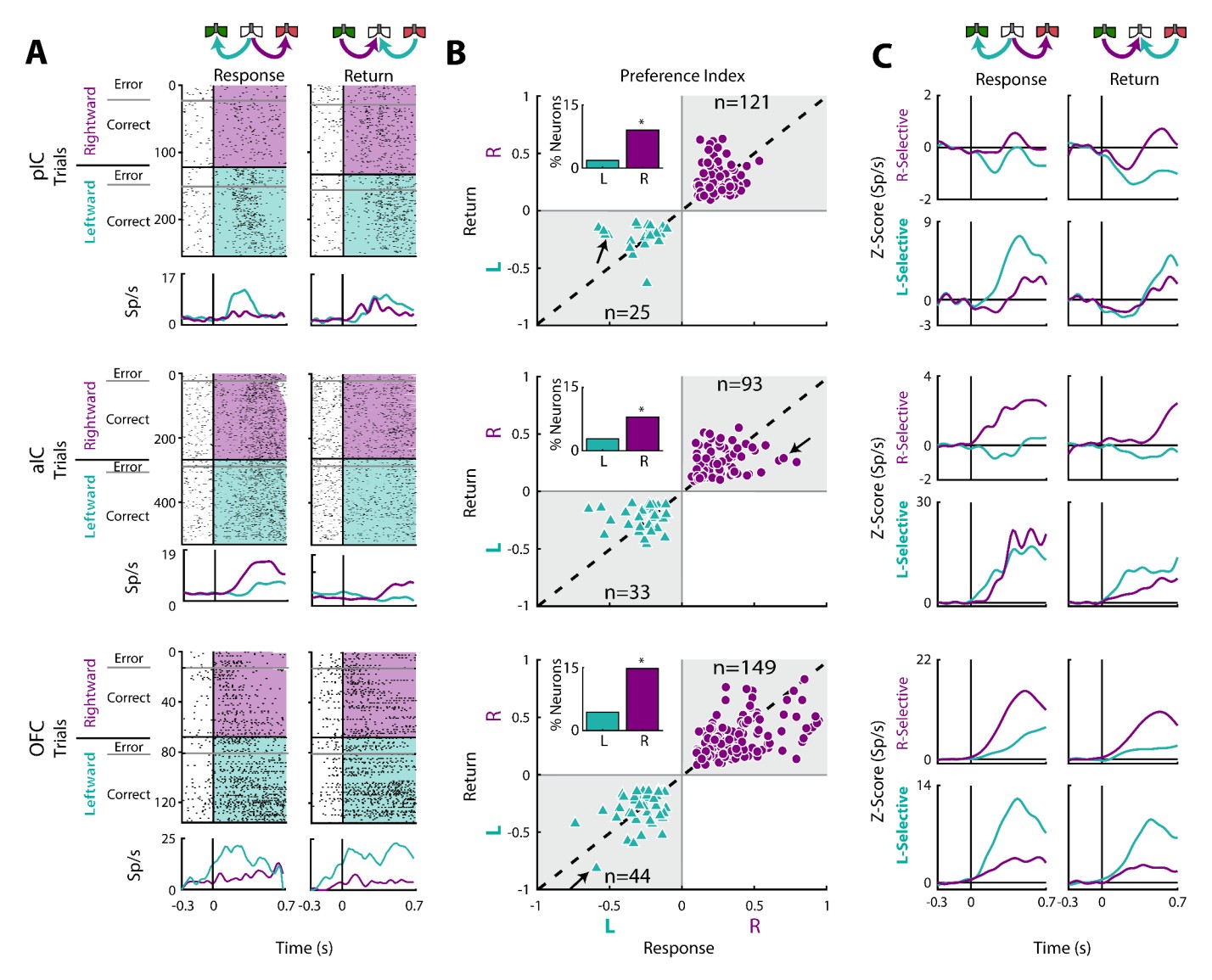

**Figure 6.** Movement direction coding in pIC, aIC, and OFC. (**A**) Three representative examples of movement direction neuronal responses in the pIC, aIC, and the OFC are depicted in the upper, middle and lower panel, respectively. In the rasters, each row is a single trial aligned to the beginning of the Response (left) and Return (right) epochs. Trials were sorted according to the movement direction and task performance: rightward (purple) and leftward (cyan) and error and correct trials. Each black tick represents a single spike. Below the rasters are displayed its corresponding PSTHs. The error trials were omitted for clarity. (**B**) Preference Indices for left (−1) or right (+1) side during the Return vs. the Response epoch (see Materials and methods). The quadrants in gray indicate where the Rightward- (purple circles) and Leftward-selective (cyan triangles) neurons are expected to be. The Inset depicts the proportion of neurons with Left/Right selectivity. The black arrows signal the examples displayed in panels A. (**C**) Normalized firing rates for Rightward- (upper) and Leftward-selective (lower panel) neurons, during the Response (left) and Return (right side) epochs. Data are expressed as mean ± sem. It is seen that the OFC exhibits the greatest difference in the Z-scores in comparison to pIC and aIC.
DOI: https://doi.org/10.7554/eLife.41152.020

*Figure 7C* depicts the lick rates during reward delivery (dashed line) and omission (solid line). Note that the rats rapidly detected reward omission since they stopped licking faster when water was omitted (the arrows indicate the second rewarded lick after delivery of the first water reward - time = 0 s- for RW trials; also see *Figure 1H*).

Finally, to quantify, the temporal dynamics of behavioral and neural decoding of the outcome, we ran a population decoder analysis. The data presented in *Figure 7D* revealed that pIC, aIC, and OFC contain neurons that detect and provide more information about reward omission than licking

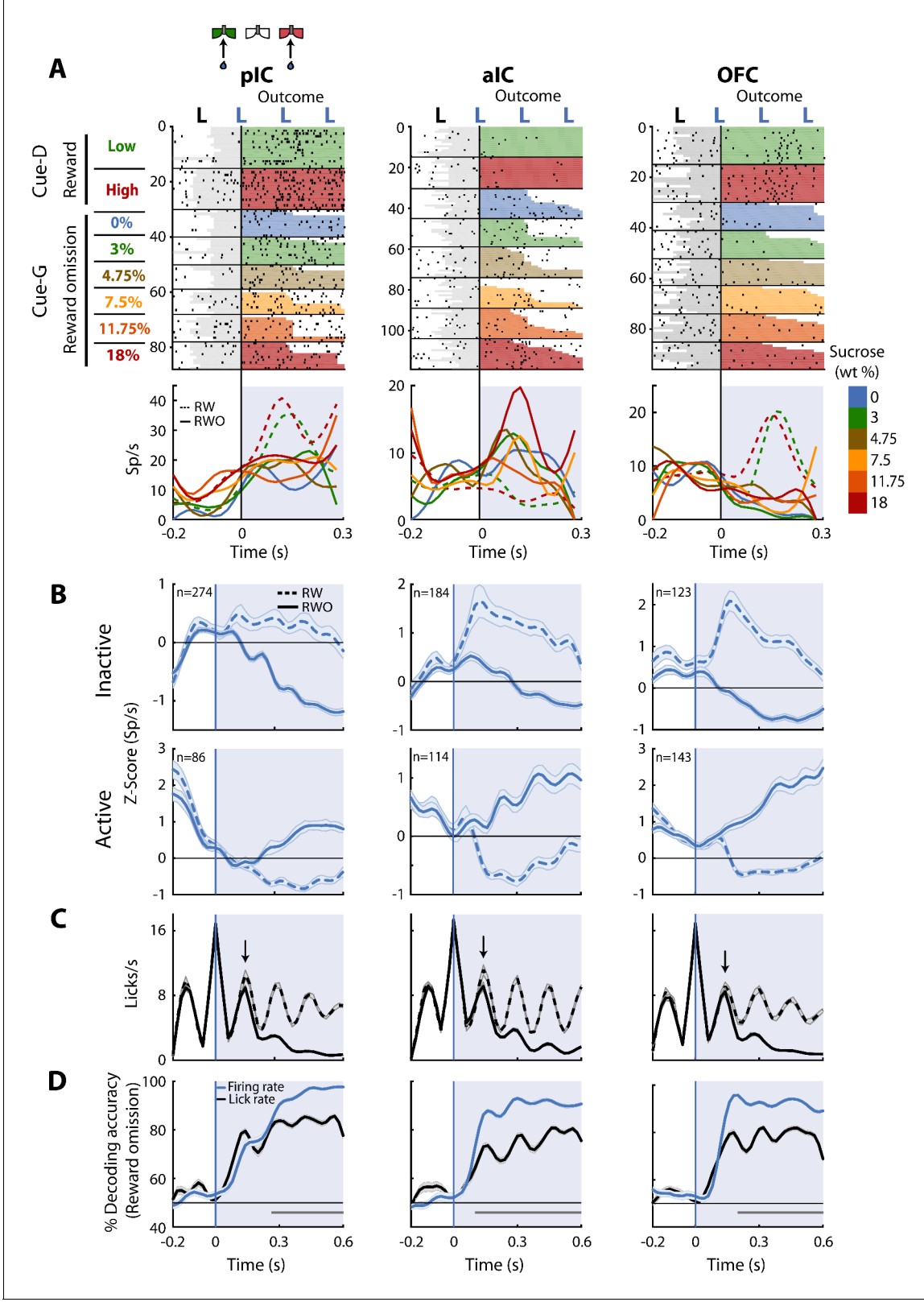

**Figure 7.** Neurons in pIC, aIC, and OFC are sensitive to reward omission. (**A**) Shown are three representative neuronal responses from the pIC (left panel), aIC (middle panel), and OFC (right panel) that encoded reward omission (RWO). The raster plot was aligned to water delivery upon the second lick in a goal-port lateral spout. The first lick was always dry. In the raster plots, the first two rows are for the rewarded (RW) Low (3%) and High (18%) correct discrimination trials. The sessions below were for the six types of generalization trials that were not rewarded (RWO; Cue-G trials). The sucrose

*Figure 7 continued on next page*

*Figure 7 continued*

concentration is indicated by the color-coded bar on the right side. Below are the PSTHs for Cue-D discrimination trials (dashed lines) and Cue-G generalization trials (solid lines). The blue-shaded rectangle indicates responses in Outcome epoch. (B) Population activity (Z-score) of the Inactive (upper panel) and the Active Reward-omission population (lower panel). These reflect those responses that either decreased or increased their firing rates after reward omission (continuous blue lines) relative to reward delivery (blue dashed line). (C) Lick rates from all generalization sessions. Continuous and dashed black lines indicate lick rates during reward omission and rewarded trials, respectively. The population PSTHs, of the firing and lick rate, were expanded from −0.2 to 0.6 s from the second lick to better appreciate the difference in firing and lick rates elicited by outcome delivery or omission. Note that the subjects required only one additional lick to detect reward absence (see arrows). (D) Decoding accuracy of the population of Outcome neurons discriminating between rewarded and unrewarded trials, using either the firing rates (blue) or the lick rates (black). The horizontal dark-gray line depicts where differences reached statistical significance.

DOI: https://doi.org/10.7554/eLife.41152.021

behavior itself. That is, the decoding accuracy was better when the algorithm used spiking activity (blue line) instead of the licking rates (black line). Our results suggest that all three of these cortical taste regions are highly sensitive to both reward delivery and reward omission.

## Integration and overlap among coding profile

Given that neurons encoding sensory and decision-variables were detected in different task epochs, we tested if there were any overlapping populations. This was accomplished using a Fisher's exact test to determine if the proportion of neurons that belong to two coding categories was above-expected chance levels. *Figure 8A* depicts a contingency table of the pairwise comparison of each coding profile category. For example, the left and middle quadrants indicates the number of neurons that encodes both Sensory and Direction (the parenthesis indicates the corresponding percentage of overlapping). Also, since in the discrimination task the Low and High Cue-D were also associated with a left/right movement, it is possible that some Sensory responses recorded in the Stimulus epoch, besides discriminating sucrose intensity, could jointly encode movement direction (in the Response epoch). We reasoned that if this were the case, then most Sensory neurons will also belong to the Direction population. This overlap was significant only in pIC (*Figure 8A*, upper panel; in 10% of the neurons). Moreover, the same result was found when we used all the Intensity-selective neurons to compute the contingency matrix (data not shown). Thus, it is unlikely that most Sensory neurons (or Intensity-selective neurons) jointly encoded sucrose intensity and movement direction. Instead, the results suggest that Sensory neurons play a more circumscribed role in chemosensory sucrose intensity processing.

Other observations revealed that the OFC Direction population was significantly associated with Choice and Outcome (*Figure 8A*, lower panel), suggesting that OFC neurons are capable of carrying, at multiple time periods, more than one spatiomotor variable related to performing the discrimination task.

## The overlap between modulation and coding profiles

We also explored if neurons encoding decision-variables (coding profile) tend to exhibit a specific modulation profile (i.e., Phasic, Tonic, Coherent). It is important to note that all modulation profiles could, in principle, encode almost any of the sensory and decision-variables. However, only a few subpopulations exhibited a significant overlap. In general, no systematic overlapping pattern was shared across the three cortical regions, suggesting that by knowing the modulation profile of one neuron provides little, if any, information about what kind of decision-variables it might encode. That said, we observed that Lick-coherent neurons in the pIC and aIC had a higher likelihood of encoding decision-variables, except for Choice neurons in the aIC which were non-preferentially encoded by any modulation profile (*Figure 8B*, upper and middle panels). In contrast, Tonic-Active neurons in the OFC jointly encoded Choice, Direction, and Outcome variables (*Figure 8B*, lower panel). In sum, these data suggest a prominent role of the Lick-coherent neurons in encoding critical features of the task in the Insula, whereas in the OFC the tonic activity prevails in encoding decision-variables.

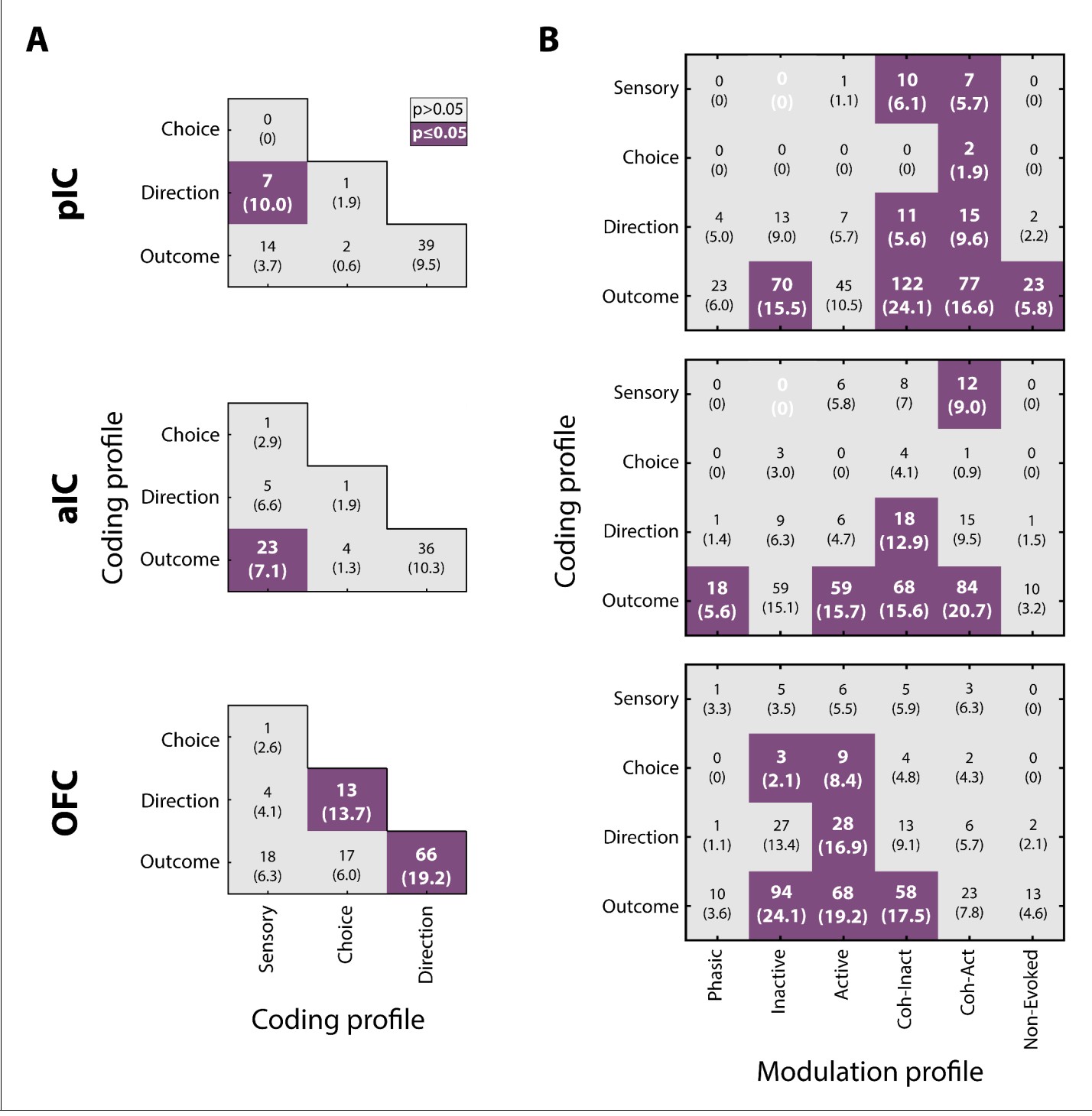

**Figure 8.** Neurons in pIC, aIC, and OFC encode more than one task-related variable. (**A**) Overlap of coding profiles. Contingency matrix indicating the number (and percentage) of neurons that belong to more than one group. Purple squares depict significant overlap as detected by a Fisher's exact test (p<0.05). (**B**) Contingency matrix showing the overlap between coding and modulation profiles. Same conventions as in A. Only neurons recorded from generalization sessions were included in this analysis to guarantee that data is drawn from the same distribution. Non-significant associated categories are indicated with a white 0. Data are presented as a number of neurons and percent in parenthesis.

DOI: https://doi.org/10.7554/eLife.41152.022

## Discussion

Sucrose intensity is a sensory attribute that contributes to the overconsumption of high-energy palatable foods (*Avena et al., 2008*; *Spector and Smith, 1984*; *Veldhuizen et al., 2017*). This study was undertaken to uncover how the perceived intensity of sucrose is represented across rat taste-related cortices (pIC, aIC, and OFC) and how this representation is transformed into decision-making variables such as choice or movement direction. We found that most neurons in these areas were responsive to the introduction of sucrose in the mouth. However, only a small subpopulation of them, in all areas, exhibited responses that tracked sucrose concentrations and that decoded sucrose's intensity equally well. Further analysis revealed that information about sucrose's intensity was conveyed in the both neuronal firing rate and spike timing. We also identified a population of neurons that tracked sucrose concentrations (Sensory neurons) with the ones that increased reflecting changes in sucrose's intensity and the ones that decreased possibly reflecting changes involving either osmotic pressure (*Hanamori, 2001*; *Lyall et al., 1999*) or the washing out of bicarbonate ions in saliva (*Zocchi et al., 2017*). In addition, neurons in the pIC and aIC encoded movement direction although OFC neurons tracked direction better than those in the Insula. Also, the neuronal signals related to Outcome (reward) were tracked similarly by these three cortical regions. In sum, we found that, in rats at least, a small and distributed group of Intensity-selective neurons represent sucrose's intensity, whereas decision-variables were also encoded in a distributed manner, but the OFC tends to encode choice and movement direction earlier and better than the Insula. These findings add to our understanding of the neural representation of sucrose's intensity in these three taste cortices and contribute to the elucidation of the decision-making processes that underlie choices guided by the concentration of sucrose.

### Cortical representation of the perceived intensity of sucrose

The neural representation of sweet taste intensity has been usually characterized by firing rates that monotonically (or sigmoidally) increase with sucrose concentration along the gustatory axis from the periphery to taste cortices (*Barretto et al., 2015*; *Rolls et al., 1990*; *Roussin et al., 2012*; *Scott et al., 1991*; *Thorpe et al., 1983*; *Villavicencio et al., 2018*; *Wu et al., 2015*). These experiments usually have been performed in animals that do not have to make any other decision than to lick (*Stapleton et al., 2006*; *Villavicencio et al., 2018*) or have the tastant passively delivered (*Katz et al., 2002*). However, as noted, the intensity attribute of a tastant can only be measured in behaving animals that actively report the perceived concentrations of sucrose. To address this issue, we developed a sucrose intensity discrimination task (see *Figure 1*) while recording from three cortical taste areas. We found that ~95% of recorded neurons were responsive to a single drop of sucrose (Cue-D), but the majority of them were unable to distinguish between 3 and 18 wt% sucrose. We posit that such a massive number of responsive neurons, which includes Intensity-selective and Non-selective neurons (see *Figure 3—figure supplement 1*), could be the result of the arrival of a salient cue (Cue-D) to an over trained animal. This stimulus is highly relevant in the context of thirsty subjects whose internal state would motivate them to attend to the delivery of an stimuli whose accurate detection and identification will lead to obtaining water. These findings are in agreement with observations that a state of physiological need (e.g., hunger) gates insular cortex responses to food cues (*Livneh et al., 2017*) and that caudolateral OFC neurons are sensitive to hunger (*Rolls et al., 1989*). Likewise, in head-fixed trained mice, it was recently reported that odor stimulation also triggers a massive widespread cortical activation in mice performing a Go/No-Go goal-directed behavior (*Allen et al., 2017*).

Multiple pieces of evidence support the idea that a small population of cortical neurons could represent sucrose intensity. For example, electrophysiological studies in rodents and non-human primates have reported a low proportion (ranging between 2–35%) of Insular (pIC) and OFC neurons with selective responses to at least one taste quality (*Pritchard et al., 2005*; *Rolls et al., 1990*; *Rolls et al., 1989*; *Scott et al., 1991*; *Stapleton et al., 2006*; *Thorpe et al., 1983*; *Yamamoto et al., 1989*; *Yaxley et al., 1990*). In contrast, one recent study in anesthetized mice, using a calcium sensor (GcAMP6), reported almost 90% of taste responses in the pIC were tastant selective although only 26% were sucrose-best (*Fletcher et al., 2017*). The differences in proportions could be explained by different data analysis and experimental preparations employed (*Chen et al., 2011*; *Katz et al., 2001*). Unfortunately, only a few studies have reported the proportion of cortical

neurons tracking taste intensity. In this regard, in the monkey Insula cortex, *Scott et al. (1991)* found that less than 1.5% (24/1661) of the recorded neurons responded linearly to increasing glucose concentrations. To the best of our knowledge, our data is the first demonstration that a small subset of cortical neurons represents sucrose intensity better than the entire population. Although intensity-selective neurons comprise a 'small' population relative to Non-selective neurons (*Figure 3B* and *Table 2*), we note that 18% of neurons in a rat's cortex would represent a large population. Further studies should investigate whether Intensity-selective neurons are sucrose-selective or broadly tuned (*Erickson, 2001*).

In this regard, it is interesting that Intensity-selective neurons were present in the recorded three cortical regions. This result is somewhat surprising in light of the experiments in the mouse Insula cortex showing the existence of non-overlapping posterior (aversive-pIC) and anterior (appetitive-aIC) 'hotspots' (*Chen et al., 2011*). Gain and loss of function experiments in mice have demonstrated that the 'sweet hotspot' is sufficient and necessary for sweet taste recognition (*Peng et al., 2015*), and a similar topographic separation of disgust-appetitive in monkeys' anterior Insula has been found, although in a different anatomical axis (dorsal–appetitive and ventral-aversive (*Jezzini et al., 2012*)). One explanation for the distributed responses that we observed (*Figures 2* and *3*) is that sucrose's identity is initially encoded in the sweet hotspot (located in aIC), but information about its perceived intensity is then distributed to other areas. Further experiments should involve the inactivation of one or more of these areas. Nevertheless, we found that each recorded cortical region decoded sucrose intensity equally well, including the pIC which according to (*Chen et al., 2011*) is where the 'aversive hotspot' is located. Thus, whatever the explanation, each of these three cortical regions contains information about sucrose intensity, revealing the distributed nature of taste intensity coding. However, the fact that all three areas decoded sucrose intensity equally well does not imply that they represent the same information, but rather they may encode different features of the sucrose intensity cues. In this regard, and despite that stimulation of the pIC elicits aversive behavioral responses (*Peng et al., 2015*), we posit that the pIC should also plays a general role in gustation since it receives most inputs from the gustatory thalamus (*Cechetto and Saper, 1987*), which could rationalize why there is sucrose responses in this cortical area (*Fletcher et al., 2017*). The aIC responses might be related to encoding the sweet percept, due to the 'sweet' hotspot, mentioned above, whose activation leads to appetitive behaviors (*Chen et al., 2011*; *Peng et al., 2015*; *Wang et al., 2018*). Finally, the OFC responses during the Stimulus epoch could also signal the relative reward value of Low and High sucrose cues (*Rolls et al., 1990*; *Tremblay and Schultz, 1999*). Our data suggest that the perceived intensity of sucrose is spatially distributed along taste cortices with a compact and distributed neural code, in the sense that a small subset of spatially disperse neurons contain more information, about sucrose intensity, than the entire population (*Field, 1994*; *Olshausen and Field, 2004*; *Stüttgen et al., 2011*).

The contribution of spike timing and spike count in taste identity coding has been extensively studied by Di Lorenzo and colleagues (*Di Lorenzo et al., 2009*; *Di Lorenzo and Victor, 2003*; *Roussin et al., 2012*). However, less is known about its contribution to the encoding of sweet intensity. In this regard, we found that additional information about sucrose's intensity was conveyed in the spike timing of neurons (*Figure 3B*). A recent study in the olfactory system reported that piriform cortex neurons encode odor intensity by using only the spike timing, and not the spike count information (*Bolding and Franks, 2017*). Likewise, our results revealed that spike timing carries additional information about taste intensity. However, spike count is also a contributor since we found Sensory neurons that tracked the concentration of sucrose by increasing its firing rate (*Figure 4*). Thus, in the taste system, it seems that both spike count and spike timing information could be complementary codes for the perceived intensity of sucrose.

Precise spike timing entrained by rhythmic licking serves as an internal clock, relevant for coordinating activity across brain regions (*Gutierrez et al., 2010*; *Gutierrez et al., 2006*; *Roussin et al., 2012*). We found that the Lick-Coherent neurons were both sufficient and necessary to decode the perceived intensity of sucrose (Tables 3 and 4). More importantly, we also uncovered, for the first time, that the level of coherence was significantly higher in the Stimulus-epoch in comparison with the pre-Stimulus and the Outcome epochs (*Figure 3—figure supplement 3*). This result implies that lick-spike coherence not only reflects oromotor responses, but that it is also involved in gating the input of sensory and taste information that can be 'read out' across taste cortices in coordination with licking (*Buzsáki, 2010*; *Gutierrez et al., 2010*).

## Decision-making in taste cortices guided by sucrose concentration

The best way to access the representation of the perceived intensity of sucrose is by allowing animals to make a decision about its intensity. Thus, it is important to determine the extent to which neuronal activity correlates with the animal's behavioral choices. We found that a distinct subset of neurons exhibited Choice-related activity in aIC and OFC with responses that covaried with the subject's choices (*Figure 5*). Furthermore, OFC (but not aIC) neurons tracked choice before a response was emitted. These findings are in agreement with behavioral observations which suggests the subjects have already made a decision before leaving the central port (*Perez et al., 2013*; *Uchida and Mainen, 2003*). Our findings reveal a neural correlate of the perceived intensity of sucrose in the gustatory system.

## Encoding of movement direction in taste cortices

Spatial navigation is an essential behavior that allows organisms to explore the environment and direct their actions toward a goal (*Epstein et al., 2017*). Spatial variables such as direction are essential to reach the desired outcome or to avoid punishment. Although spatial information is encoded in brain regions specialized for spatial processing, such as the hippocampus and entorhinal cortex, recently it has been found that other unexpected areas also contain spatial information (*Yin et al., 2018*). In this regard, here we also found that OFC neurons robustly encoded movement direction. Likewise, neurons with direction selectivity in the OFC have been recorded in tasks involving two or four spatial locations (*Feierstein et al., 2006*; *Lipton et al., 1999*; *Roesch et al., 2006*). Lesioning the OFC disrupts performance in an allocentric foraging task (*Corwin et al., 1994*), and radial arm and Morris water maze (*Kolb et al., 1983*). Moreover, the OFC also encodes head angle, spatial trajectory and movement speed in a spatial discrimination and reversal task in a plus maze (*Riceberg and Shapiro, 2017*). The latter evidence agrees with the high proportion of OFC Direction-selective neurons that we identified (*Figure 6*). In contrast, less is known about the participation of the Insular Cortex in encoding spatial navigation parameters; although, it is known that ablating either the pIC or the aIC results in a severe impairment of spatial navigation in a water maze (*Nerad et al., 1996*). Here, for the first time, we found that neurons in the pIC and aIC tracked movement direction, probably through their connections with the entorhinal cortex (*Wang et al., 2018*). However, according to the $P_{index}$ values, the encoding of direction was weaker in the IC in comparison to the OFC (*Figure 6*). Altogether, our data points to a dominant role for the OFC, and to a lesser extent the IC, in encoding movement direction; an essential feature of spatial navigation for goal-directed behaviors.

The detection of either reward delivery or reward omission is essential for animals' survival and for triggering learning based on reward prediction errors (*Schultz et al., 1997*). Previous observations have shown that aIC and OFC neurons encode reward omission (*Feierstein et al., 2006*; *Jo and Jung, 2016*) and we found the pIC, aIC, and the OFC differentially respond to the presence and absence of reward (*Figure 7*); suggesting a distributed tracking of reward omission. However, this is the first demonstration that pIC neurons could also encode reward omission. The pIC has a key role in updating the current outcome representation to guide action selection. This is because without affecting the execution of the instrumental responses its chemogenetic inhibition impairs the ability of subjects to adjust their actions based upon the outcome current value (*Parkes et al., 2018*; *Parkes et al., 2015*). Our results demonstrate a widespread representation of neural signals related to the Outcome, which is a crucial process for learning and adaptive behavior.

As noted above, we identified several differences and similarities between the evoked pIC, aIC, and OFC responses. The main similarity among all three brain regions was that they decoded sucrose concentration equally well. A major difference was that OFC neurons carry information about decision-variables earlier and with higher quality than neurons in the Insula (*Figures 5–7*). That is, unlike the Insula, the OFC was the brain region with more neurons jointly encoding more than one decision-variable (Choice, Direction, and Outcome; *Figure 8A*), indicating that the OFC has a complete representation of the most relevant task events. It follows that the OFC provides an up-to-date representation of task-related information that is required to yield the best outcome. In reinforcement learning, this information is named 'state' representation (*Schuck et al., 2018*; *Stalnaker et al., 2016*; *Sutton and Barton, 1998*).

The OFC is also involved in encoding the subjective reward value of associated choices (*Conen and Padoa-Schioppa, 2015*; *Rolls, 2004*; *Tremblay and Schultz, 1999*). However, in our task correct actions (choosing left/right) led to the same reward (i.e., 3 drops of water), suggesting, in agreement with findings in an odor guided task (*Feierstein et al., 2006*), that OFC neurons could encode spatiomotor variables, such as Choice and Movement direction, even for actions with the same reward value. Our results both confirm and extend these findings by further demonstrating that OFC neurons could represent decision-variables in a task guided by the intensity of sucrose. We posit that OFC may act as a hub that represents decision-variables regardless of the type of sensory input used to guide goal-directed behaviors. The OFC is a brain area well suited to perform this function since it receives connections from sensory areas related to olfactory, gustatory, visual, and somatosensory processing (*Cavada et al., 2000*).

### Concluding remark

We found evidence that in animals trained to identify sucrose intensity the taste system uses a compact and distributed code to represent its perceived intensity. Moreover, the perceived intensity of sucrose and the decision-variables associated with the discrimination task can be fully reconstructed from a small population of neurons in the pIC, aIC, and OFC.

## Materials and methods

### Chemicals

Sucrose was reagent-grade chemical quality purchased from Sigma-Aldrich (Mexico City, Mexico). It was dissolved in distilled water and used the following concentrations 3, 4.75, 7.5, 11.75, and 18 wt/vol%. Solutions were prepared fresh every other day. They were maintained under refrigeration, and they were used at room temperature.

### Subjects

We used 28 male Sprague-Dawley rats weighing 300–320 g at the beginning of the experiment, and by the end of recordings, their weights were 412.3 ± 8 g. Animals were individually housed in standard laboratory cages in a temperature-controlled (22 ± 1°C) room with a 12:12 h light-dark cycle (lights were on 0700 and off at 1900). All procedures were approved by the CINVESTAV Institutional Animal Care and Use Committee. During experiments, rats were given *ad libitum* access to tap water for 30 min after testing. Chow food (PicoLab Rodent Diet 20, St. Louis, MO, USA) was always available in their homecage. All experiments were performed in the late light period from 1400 to 1900 h since at this period rats were more alert and motivated to work.

### Behavioral equipment

Animals were trained in four identical standard operant conditioning chambers of internal dimensions 30.5 × 24.1×21.0 cm (Med Associates Inc, VT, USA). The front panel of each chamber was equipped with one central and two laterals V-shaped licking ports with a photobeam sensor to register individual licks (Med Associates Inc, VT, USA). Each port had a licking spout that consisted of either one (for lateral ports) or a bundle of up to 6 (for the central port) blunted needles (20-gauge) that were carefully sanded and glued at the tip of a stainless-steel sipper tube. Each needle was connected to a solenoid valve (Parker, Ohio, USA) via a silicon tube. The volume of the drop was adjusted before each session and maintained by using an individual and constant air pressure system (*Perez et al., 2013*). On the rear panel, there was an ambiance white noise amplifier with a speaker that was turned on in all the sessions. Chambers were enclosed in a ventilated sound-attenuating cubicle. Experimental events were controlled and registered by a computer via a Med Associates interface (Med Associates Inc, VT, USA).

### Sucrose intensity discrimination task

All subjects were trained in a 'Yes/No' psychophysical task (*Stüttgen et al., 2011*) to emit a response by either going left or right based on the concentration of a 10 µL sucrose cue (Low 3% or High 18 wt%). For trained animals, the task comprises four epochs: Return, Stimulus, Response, and Outcome. The outline of a trial is depicted in *Figure 1A*. A trial began when trained subjects moved

from either lateral port to return the central spout; this epoch was named Return. Once in the central port, the rats were required to lick the empty spout a variable number of times (between two or three) to receive a 10 μL drop of either 3 or 18 wt% sucrose (hereafter Cue-D). Rats could give additional empty licks after Cue-D delivery. These empty licks were used as a measure of the palatability oromotor responses elicited by sucrose (*Perez et al., 2013*). The time elapsed from Cue-D delivery to the last lick in the central spout was designated as the Stimulus epoch. Subsequently, subjects had to move to either the Low or High sucrose-associated port (Response epoch) and emit, at least, one dry lick. If the response was correct, subsequent licks delivered three drops of water as a reward, while incorrect choices briefly turned off and on the lights during 50 ms (at the second dry lick) and subsequent licks were without a reward. The Outcome port comprises the interval where rats were licking in the lateral spout. The learning criterion was set at ≥80% correct responses during four consecutive sessions.

Importantly, a drop of water was not delivered at the central port as a washout because in a pilot study we found that rats did not learn the task despite extensive training (>50 sessions). We speculate that this was due to an imbalance in the reward value between the licking ports. Specifically, the reward value of one drop of water +one drop of sucrose (3 or 18 wt%) at the central spout seems to be higher than the value of 3 drops of water delivered at the lateral spouts. The inclusion of a water washout failed to motivate rats and induced learning and thus the water washout, at central spout, was no longer used.

## Generalization sessions

Once the animals learned to discriminate between Low (3 wt%) and High (18 wt%) sucrose by getting at least 80% of the trials correct, the generalization sessions were introduced. Generalization sessions were composed of 20% of the trials (80% were of discrimination trials). These trials were like discrimination trials with the exception that after at least two discrimination trials subjects received a drop of either 0, 3, 4.75, 7.5, 11.75, or 18 wt% sucrose. In these trials, no reward was delivered after choosing either lateral spout (and in the second dry lick, the lights turned briefly on and off for 50 ms, signaling that no reward will be delivered). Discrimination and generalization sessions were interleaved, such that a generalization session occurred if at least one discrimination session with ≥80% correct responses took place the day before. This procedure avoids impairment of task performance.

Since no statistical differences in task performance were found among groups, behavioral data were collapsed across subjects for the three brain regions recorded. For discrimination sessions, the percent correct responses were obtained by counting the number of trials for Low or High that subjects responded to the correct associated choice spout, divided by the total number of trials. To determine if performance was affected by electrode implantation, the average performance of the five sessions pre- and post-surgery were compared using a paired t-test (*Figure 1*). For generalization sessions, the percent responses given to the High concentration spout was plotted, and a sigmoid function was fitted to obtain the psychometric function. Likewise, surgery effects over generalization sessions were evaluated by comparing the average performance for all these sessions before and after surgery with a paired t-test. The time spent licking in the central port for each concentration (Cue-D +Cue G trials) during generalization sessions were collapsed and compared using a one-way ANOVA, and a Dunnett *post hoc* confirmed differences against sucrose 3 wt% Cue-D trials. As well, the time spent during the Return and Response epochs for each movement direction (left or right), and during the Outcome epoch for reinforced and unreinforced trials, were collapsed and compared using an unpaired t-test.

## Surgery and histology

Once animals achieved the learning criterion and at least three consecutive generalization sessions were tested, then we proceeded to implant a custom-made 16 tungsten wires (35 μm diameter) each arranged in a 4 × 4 (1 mm$^2$) multielectrode array. The array was implanted in the posterior Insula (pIC; n = 11), in the anterior Insula (aIC; n = 8) and the orbitofrontal cortex (OFC, n = 9). All subjects were anesthetized using ketamine (70 mg/kg, i.p.) and xylazine (20 mg/kg, i.p.). The rats were put in a stereotaxic apparatus where a midline sagittal scalp incision was made to expose the skull and to put two holding screws. A third screw soldered to a silver wire that served as an

electrical ground for recordings was inserted above the cerebellum (*Gutierrez et al., 2010*). A craniotomy in the left hemisphere was made to implant an electrode array in one of the following sites: posterior IC (AP: +1.0 to +1.4 mm, ML: +5.2 mm from bregma, DV: −4.4 to −4.7 mm ventral to dura), anterior IC (AP: +1.6 to +2.3 mm, ML: +5.2 mm from bregma, DV: −4.6 to −4.7 mm ventral to dura) or OFC (AP: +3.5 mm, ML: +3.2 mm from bregma; DV: −4.4 mm ventral to dura). Dental acrylic was applied to cement the electrode array to the screws. The rats were given intraperitoneal enrofloxacin (0.4 ml/kg) and ketoprofen (45 mg/kg) for three days after surgery and were allowed to recover for one week. After the completion of the experiments, subjects were deeply anesthetized with an overdose of pentobarbital sodium (150 kg/mg, i.p.) where they were transcardially perfused with PBS (1x) followed by 4% paraformaldehyde. Brains were removed, stored for one day in 4% paraformaldehyde and posteriorly were changed to a 30 vol./vol.% sucrose/PBS solution. Brains were sectioned in 40 μm coronal slices, and they were stained with cresyl violet to visualize the location of electrode tips.

## Electrophysiology

Neural activity was recorded using a Multichannel Acquisition Processor system (Plexon, Dallas, TX) interfaced with a Med Associates conditioning chamber to record behavioral events simultaneously. Extracellular voltage signals were first amplified x1 by an analog headstage (Plexon HST/16o25-GEN2- 18P-2GP-G1), then amplified (x1000) and sampled at 40 kHz. Raw signals were band-pass filtered from 154 Hz to 8.8 kHz and digitalized at 12 bits resolution. Only single neurons with action potentials with a signal-to-noise ratio of ≥3:1 were analyzed (*Gutierrez et al., 2010*). The action potentials were isolated on-line using voltage-time threshold windows and three principal components contour templates algorithm. A cluster of waveforms was assigned to a single unit if two criteria were met: Inter-Spike Intervals were larger than the refractory period set to 1 ms, and if it is formed a visible ellipsoid cloud composed of the 3-D projections of the first three principal component analysis of spike waveform shapes. Spikes were sorted using Offline Sorter software (Plexon, Dallas, TX) (*Gutierrez et al., 2010*). Only time stamps from offline-sorted waveforms were analyzed.

## Data analysis

All data analysis was performed using MATLAB (The MathWorks Inc., Natick, MA) and Graphpad Prism (La Jolla, CA, USA). Unless otherwise indicated, we used the mean ± sem and the α level at 0.05.

## Modulation profiles

### Cue-evoked responses

In the Stimulus epoch, five major Cue-evoked responses were identified: Phasic, Tonic either Active or Inactive and Lick-coherent either Active or Inactive (*Figure 2*). We compared the proportions of Cue-evoked responses among brain regions using a chi-square test. Only correct trials were analyzed. Each Cue-evoked response type fulfilled a criterion that is described in detail below:

### Phasic responses

To determine if there were phasic Cue-evoked responses, we compared the firing rate from 0 to 0.2 s after Cue-D delivery against the baseline, which encompassed the dry licks emitted from −0.3 to onset of Cue-D delivery (*Villavicencio et al., 2018*). If there was a significant difference (Wilcoxon rank-sum test), then we used a cumulative sum test to identify the onset of modulations (see *Gutierrez et al., 2006*). This analysis identified the onset and offset of modulations by detecting in which time bin the firing rate significantly increased or decreased relative to the baseline (*Figure 2— figure supplement 1*). A neural response was denominated as Phasic if there was one excitatory modulation that started between the first 0.1 s after Cue-D delivery and the duration of this modulation was within 0.04 and 0.2 s. This procedure assures that only phasic (but not tonic) modulations were selected.

### Tonically-Inactive and -Active responses

To determine whether a neuron showed a statistically significant evoked response during the Stimulus epoch, we used a 'best window' analysis. The analysis consisted of scanning the firing rate after

Cue-D delivery in multiple window sizes (from 0.05 to 0.6 s in steps of 0.05 s) encompassing only the interval from 0 to 0.6 s. Hence, the firing rate in a variety of time centers (from 0.05 to 0.5 s, in 0.05 s steps) and for multiple window sizes was computed, such that each window was estimated as the center ± (window size/2). The firing rate on each window was compared against baseline using a Wilcoxon rank-sum test. For all statistically significant windows, the 'best window' was the one with the largest delta (change) in firing rate relative to baseline. A modulation was assigned to be tonic if the duration of the modulation was greater or equal to 0.2 s. Positive modulations were termed Active, while negative modulations were designated as Inactive.

### Lick coherent responses (Coh-Inactive and -Active)

Oscillatory activity between spikes and rhythmic licking (in the 4–12 Hz bandwidth) was identified using multi-taper spectral analysis (*Jarvis and Mitra, 2001*) by segmenting into chunks the PSTHs aligned to the first lick given at the central spout (for additional details see *Gutierrez et al., 2010*). The confidence intervals and the significance threshold were determined by using a jackknife method (*Jarvis and Mitra, 2001*). A neuron was classified as Lick-coherent only if the lower confidence interval was above the significance threshold. To detect if a Lick-coherent neuron exhibited a Cue-D-evoked response a 'best window' analysis was employed. To determine whether the modulation was either inactive or active the mean firing rate of the significant window was subtracted from the baseline. If the result was positive, the modulation was named Lick-coherent Active, while if it was negative, it was named Lick-coherent Inactive. From these Lick-coherent neurons we calculated the average coherence value (in the 4–12 Hz band) between licks and spikes in the three brain regions. Differences in coherence values were analyzed using a one-way ANOVA, and a Tukey *post hoc*. Finally, in order to assess the relevance of coherence in the detection of gustatory cues, we calculated the coherence value during three task epochs. Specifically, we used the central licks given before (pre-Stimulus) and after Cue-D delivery (Stimulus epoch), and the reinforced licks given during the Outcome epoch. Coherence value between epochs was compared by using a one-way ANOVA, and a Tukey *post hoc* confirmed further differences.

### Non-modulated and Coherent-Non-evoked responses

Neuronal responses that displayed no significant statistical differences between baseline and cue delivery were denoted Non-Modulated. Neurons that were Lick-coherent but which firing was not significantly different between baseline and 0.6 s after Cue-D delivery were termed as Coherent Non-evoked. When Non-modulated and Coherent-Non-Evoked neurons were collapsed they were named as Non-evoked.

### Intensity-selective neurons

To determine Intensity-selective responses we also used the 'best window' approach. A Wilcoxon rank-sum test was applied to compare the firing rate between Low vs. High Cue-D trials during different intervals within the evoked response. The significant window with the higher delta in the firing rate between intensities was named the 'best window.' Neurons that responded similarly to both intensity cues were named Non-selective. A chi-square test was used to determine differences in the proportion of Intensity-selective and Non-selective neurons among cortical regions.

To measure the differences in licking and its impact on neural responses associated with sucrose's intensity, we determined if lick rate differences occurred during the best-window of Intensity-selective neurons and, if so, we then quantified the proportion of the best-window with lick rate differences. To accomplish this, we employed a Receiver Operating Characteristic (ROC) analysis (*Green and Swets, 1966*) that quantified how different were the lick rates distributions of Low and High trials during the first 0.6 s of the Stimulus epoch. The area under the ROC curve (auROC) and its confidence intervals were calculated for all bins (bin size 0.1 s). If the inferior confidence interval of a specific bin was above 0.5, then lick rates were significantly different in that bin. Only Intensity-selective with significative lick rate differences within the best-window were counted and considered to calculate the proportion of the best-window that displayed lick rate differences.

## Population decoding of sucrose intensity

To evaluate if Intensity-selective neurons contained more information about sucrose intensity than Non-selective neurons we employed a neural population decoder (MATLAB toolbox of the 1.0 version of the Neural Decoding Toolbox, www.readout.info) (*Meyers, 2013*). To achieve this goal, the decoder was tested with a vector that contained the label of the sucrose intensity given (Low or High) in each trial and a matrix $m$ x $n$ that contains the number of spikes occurring in each trial ($m$) during each 20 ms time bins ($n$). This matrix is comprised of spikes occurring from Cue-D delivery 0 to 0.6 s. Then, the firing rate matrix was normalized to Z-score, and the data set was divided into $k$ different splits (number of data sets). Subsequently, $k$-1 splits were used to train the classifier by averaging the firing rate from the selected trials according to the label class (Low or High), therefore generating a Low or High activity vector where each row represents the firing rate at each time bin. The remaining split was used to test the classifier. To improve the decoder performance this procedure was repeated $k$ times using a different test split each time, the average of these results was reported as the percentage decoding accuracy. The decoder performance for Non-evoked, 'All,' Non-selective, and Intensity-selective populations were obtained. Significant differences were determined by using a one-way ANOVA and a Bonferroni *post hoc* to detect differences in performance between populations. Furthermore, to determine if spike timing conveyed information about sucrose intensity, we maintained the original firing rate but the time at which each spike occurred was shuffled, in each trial. For this, we counted the number of spikes ($n$) occurring from 0 to 0.6 s after Cue-D delivery and sampled without replacement $n$ new timestamps during this interval. A paired t-test was employed to determine differences in decoding accuracy before and after spike timing shuffle.

Since spike timing shuffling might disrupt the noise-correlation between pairs of simultaneously recorded neurons; and these correlations can affect population decoding (*Carnevale et al., 2013*; *Cohen and Kohn, 2011*; *Zohary et al., 1994*), we determined the impact of removing noise-correlations over the decoding accuracy of sucrose intensity. To do so, we detected the neuron-pairs obtained from different channels (*Cohen and Kohn, 2011*) and normalized the firing rate of each neuron relative to each stimulus (Low or High) by following this equation:

$$\frac{FR_{ksn} - <FR_{sn}>}{\sigma_{sn}}$$

Where FR is the firing rate, and $k$ indicates the trial, $s$ refers to the stimulus (Low or High), and $n$ is the neuron evaluated (Neuron A or Neuron B). Note that for each neuron the firing rate of a given intensity is normalized relative to that intensity (i.e., Low relative to Low). The normalized firing rate of Low and High trials of Neuron A were concatenated, generating one vector; the same for Neuron B. From these vectors we calculated the Pearson´s correlation coefficient of the normalized spike counts ($r_{sc}$, noise correlation); if $r_{sc}$ was significative (see *Figure 3—figure supplement 2A*, upper panel), then a permutation test where the position of trials was shuffled for Neuron A and Neuron B separately was performed in order to remove noise-correlations (see *Figure 3—figure supplement 2A*, lower panel). The procedure was repeated 10,000 times. A corrected *p*-value was obtained using the following formula: $p = (k + 1) / (n + 1)$, where $k$ is number of times that the shuffled *p* was greater than the unshuffled *p*-value and $n$ is the number of shuffling repetitions. If the corrected *p*-value was still significant, then each neuron pair was considered to be noise-correlated. This procedure was accomplished for different time windows from 0.1 to 0.6 s (0.1 s steps) after Cue-D delivery. However, since noise correlation values were similar across window sizes (see *Figure 3—figure supplement 2B*), and because 0.6 s offered the most stable firing rate across trials, we choose this window for subsequent analysis. Then, we shuffled trials positions of neurons that were noise correlated, and the decoder was fed with these shuffled matrices of the Non-evoked, All, Non-selective, and the Intensity-selective population. A paired t-test was used to compare each population before and after removal of noise-correlation.

To further explore the contribution that each Cue-evoked population adds to sucrose's decoding, an inclusion and dropping population analysis was performed by either including or removing only one population at a time, respectively (*Gutierrez et al., 2006*). For these analyses, two more populations were added: All-Tonic (combining the Tonic-Inactive + Tonic-Active) and All-Coherent (pooling Coherent-Inactive + Coherent-Active). The percent decoding accuracy for Non-evoked, 'All,' Phasic, Inactive, Active, Coherent-Inactive, Coherent-Active, All-Tonic, and All-Coherent were computed for the inclusion analysis. Differences between groups were tested with a one-way ANOVA and a

Dunnett *post hoc* to compare each group (excluding Phasic) with the Non-evoked. The first 0.2 s of the decoder performance of the Phasic group was compared against the Non-evoked group, by employing an unpaired t-test. On the other hand, for the dropped analysis the Non-evoked group was not included, and the percent accuracy when 'All' neurons were used as a control group for the Dunnett *post hoc*. As well, Phasic (0.2 s after Cue-D) was compared to All population using an unpaired t-test.

## Coding profile

### Sensory neurons

Responses that decrease or increase as a function of sucrose concentration were searched in the subpopulation of neurons that displayed Intensity-selective responses during 0.6 s after Cue-D onset (see Cue-evoked responses). We identified the 'best window' by using the firing rates in each Cue-G trial computed for different window sizes and time centers. The firing rates of each window were correlated with sucrose concentration (3, 4.75, 7.5, 11.75 and 18 wt%) using a Pearson correlation. The statistically significant window with the highest Pearson correlation coefficient was named as the 'best window.' A permutation test were intensity labels were shuffled without replacement was used as multiple-testing correction test (*Davison and Hinkley, 1997*) for the 'best window.' Briefly, data were shuffled 10,000 times. A corrected *p*-value was obtained as mention above. The Pearson correlation coefficient was computed for each significant neuron and then average for each population: increasing or decreasing activity as a function of sucrose intensity. Likewise, the activity of each population was normalized to 3 wt%, and the Z-score was computed.

### Choice-related neurons

To determine if neural activity tracked subjects' choices, we identified the 'best decision window' where the firing rate correctly classified trials as either Low or High intensity with the highest accuracy, thereby matching the responses made by the subject to the neural activity. This method has been successfully used in non-human primates to explore the somatosensory system (*de Lafuente and Romo, 2005*). The analysis was restricted to neurons recorded during generalization sessions during two window intervals: (1) from 0 to 0.6 s from Cue delivery (Stimulus epoch), and (2) from 0.4 s before to 0.8 s around Response onset (time = 0 s), covering the Stimulus and Response epoch. These interval windows were chosen to assess if neurons could track subject's decisions before they leave the central (Stimulus) port and during the Response epoch. First, using a 0.2 s fix window size and moving centers in steps of 0.1 s that encompassed all the interval window evaluated, a firing rate matrix (*m* x *n*) was obtained for Low (0, 3, 4.75 wt%) and High (7.5, 11.75, 18 wt%) trials, where *m* represented each neuron and *n* each time bin from the window interval. A ROC curve was employed to obtain the auROC index and its confidence interval following the methods described in *de Lafuente and Romo, 2005*. A neuron was considered to significantly discriminate between Low and High trials if the inferior confidence interval of at least five consecutive time bins were above a ROC value of 0.5 (no significant difference). If this criterion was met, then the 'best window' where a firing rate threshold classified a trial as being a Low or High trial, with the higher precision, was obtained. To reach this objective, the firing rate from different windows was computed and compared to different arbitrary firing rate criteria. If a neuron fired more at High trials was denominated as 'High-Preferred,' and the number of 'High trials' where the firing rate was above the arbitrary threshold were considered High Hits, while the number of 'Low trials' where firing rate was below this criterion (Low Hits) were counted. If a neuron was 'Low-Preferred' the Low and High Hits were considered trials where the firing rate was above and below the arbitrary firing rate threshold, respectively. The Low and High Hits were summed and divided by the total number of trials and converted to a percentage of correct responses. Briefly, the neuron 'responses' were calculated as follow:

$$\text{Correct neuron responses} = \frac{\text{Low Hits} + \text{High Hits}}{\text{Number of trials}} x \, 100$$

If the neurometric performance was $\geq 70\%$ correct responses, then it was considered a Choice neuron. This criterion was selected since it corresponds to the behavioral performance achieved by rats during the classification sessions (*Figure 5—figure supplement 1A*). The percentage of correct

intensity responses from all Choice neurons was then averaged and plotted against sucrose intensity and fitted to a sigmoid function to obtain the neurometric curve. The psychometric data for the sessions where these neurons were recorded was plotted and fitted to a sigmoid function. Finally, the mean and the 95% confidence interval of the slopes for the psychometric and neurometric curves were obtained. A neurometric curve was determined to reliably matched behavior only if the confidence interval of the slopes of both curves were overlapped.

## Direction-selective neurons

In the behavioral tests, subjects were required to move from the center to a lateral port (Response epoch) and from the lateral to the central port (Return epoch) to emit a response and to initiate a new trial, respectively (*Figure 1A*). A Direction-selective neuron would be more active when the subject was moving to one direction (i.e., left) during both Response and Return epochs, while when moving to the other direction (i.e., right) the neuron was less active or non-active. A ROC analysis was employed to detect differential responses for Left and Right trials from 0 to 0.7 s of the Response and Return epochs. First, the firing rates of the Left and Right trials were computed. Then, the auROC curve was determined and scaled by calculating a preference index $P_{index} = 2(auROC$ curve $-0.5)$ (*Feierstein et al., 2006*), which provides a value between $-1$ (Left preferred) and 1 (Right Preferred), where 0 meant no preference. If the neuronal response preferred one side during the Response epoch, a permutation test where left/right labels were shuffled without replacement was used. Data were shuffled 10,000 times. A corrected p-value was obtained using the following formula: $p = (k + 1) / (n + 1)$, where k is number of times that the shuffled auROC curve was greater than the non-shuffled auROC curve and n is the number of shuffling repetitions. If *p* was significant, the same analysis was repeated from 0 to 0.7 s of the Return epoch. All trials were included in the analysis since for the Direction neurons it did not matter whether it was a correct or error trial, but only the direction of the movement. The Z-score of the population activity for Left- and Right-selective neurons was computed using as baseline the 0.3 s before the central and lateral port exit for the Response and Return epoch, respectively. A chi-square test was used to determine differences in the proportion of neurons with Right- or Left-preferred Direction.

## Neurons encoding reward (outcome) omission

Neurons that were modulated by water reward omission were identified by using the same ROC analysis as described for the detection of Direction-selective neurons (see above), with the exception that this analysis was restricted to the Outcome epoch: from 0 to 0.6 s from the onset of the second lateral lick. The firing rate from rewarded (correct discrimination) and unrewarded trials (error discrimination and generalization trials) were obtained to calculate the auROC curve. The population that displayed higher firing rate for reward omitted trials was named 'Active' (positive $P_{indexes}$), while the population with lower firing rate for these types of trials was denominated 'Inactive' (negative $P_{indexes}$). Furthermore, the lick rate for each type of trials (rewarded/unrewarded) was also calculated to allow visualization of oromotor differences. The population activity was normalized to Z-score by using the firing rate in the pre-Stimulus epoch ($-0.3$ to 0) as the baseline.

## Population decoding of reward omission

The accuracy to predict the presence or absence of the reward was evaluated by providing the neural decoder with the spikes or the licks that occurred from $-0.2$ to 0.6 s around the second lateral lick. A ROC analysis was employed to detect the window intervals where performance between lick and firing rate were significantly different: if the inferior confidence interval was above 0.5 during at least five consecutive time bins (0.02 s).

## Overlapping among coding profile populations and between modulation vs. coding profile neurons

Since the neurons coding profile were identified in different epochs; then, it is possible that a neuron coding one variable (e.g., direction) could encode another variable in a different interval (e.g., outcome). Thus, we evaluated the existence of a significant overlap between coding profile populations (*Villavicencio et al., 2018*). To achieve this goal, a contingency matrix containing the number of only A, only B, A and B, non-A and non-B neurons were obtained for all pair of combinations. A Fisher's

exact test was applied to the matrices. Only neurons recorded during generalization sessions were used for the analysis to guarantee that data is drawn from the same distribution. Likewise, we determined if a coding profile subset belonged preferentially to one modulation profile by performing the same analysis: A Fisher's exact test was applied to a contingency matrix, where A was a modulation profile and B a coding profile. All possible pairwise comparisons were tested. Also, only generalization sessions were considered for the analysis.

## Acknowledgments

This project was supported in part by Productos Medix 3247, CONACyT Grants Fronteras de la Ciencia 63 (RG) and 245 (V dL), and Problemas Nacionales 464 (RG). Esmeralda Fonseca had a CONACyT doctoral fellowship and data in this work is part of her doctoral dissertation in the Posgrado en Ciencias Biomédicas of the Universidad Nacional Autónoma de México. We thank Mario Gil Moreno for building multielectrode arrays, Fabiola Hernandez Olvera for invaluable animal care, and Miguel Villavicencio for insightful comments on an early version of the manuscript. We also want to specially thank Aurora Sono Matsumoto for invaluable help training rats.

## Additional information

### Funding

| Funder | Grant reference number | Author |
| --- | --- | --- |
| Consejo Nacional de Ciencia y Tecnología | Problemas Nacionales 464 | Ranier Gutierrez |
| Productos Medix | 3247 | Ranier Gutierrez |
| Consejo Nacional de Ciencia y Tecnología | FOINS 63 | Ranier Gutierrez |
| Consejo Nacional de Ciencia y Tecnología | FOINS 245 | Victor de Lafuente |

The funders had no role in study design, data collection and interpretation, or the decision to submit the work for publication.

### Author contributions

Esmeralda Fonseca, Conceptualization, Data curation, Software, Formal analysis, Supervision, Validation, Investigation, Visualization, Methodology, Writing—original draft, Writing—review and editing; Victor de Lafuente, Software, Formal analysis, Funding acquisition, Writing—review and editing; Sidney A Simon, Conceptualization, Supervision, Validation, Visualization, Writing—original draft, Writing—review and editing; Ranier Gutierrez, Conceptualization, Resources, Software, Supervision, Funding acquisition, Validation, Visualization, Methodology, Writing—original draft, Project administration, Writing—review and editing

### Author ORCIDs

Esmeralda Fonseca https://orcid.org/0000-0003-3697-9401
Victor de Lafuente https://orcid.org/0000-0002-1047-1354
Ranier Gutierrez https://orcid.org/0000-0002-9688-0289

### Ethics

Animal experimentation: All procedures were approved by the CINVESTAV Institutional Animal Care and Use Committee (#0034-13)

### Decision letter and Author response

Decision letter https://doi.org/10.7554/eLife.41152.028
Author response https://doi.org/10.7554/eLife.41152.029

## Additional files

### Supplementary files
• Supplementary file 1. Statistical analysis for all figures.
DOI: https://doi.org/10.7554/eLife.41152.023

• Supplementary file 2. Statistical analysis for all tables.
DOI: https://doi.org/10.7554/eLife.41152.024

• Transparent reporting form
DOI: https://doi.org/10.7554/eLife.41152.025

### Data availability
All data generated or analysed during this study are included in the manuscript and supporting files

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
