## [Decision Letter]

Thank you for submitting your article "Sucrose intensity coding and decision-making in taste cortices" for consideration by *eLife*. Your article has been reviewed by three peer reviewers, including Geoffrey Schoenbaum as the Reviewing Editor and Reviewer #1, and the evaluation has been overseen by Michael Frank as the Senior Editor. The following individuals involved in review of your submission have agreed to reveal their identity: Daniel W Wesson (Reviewer #2); Yoav Livneh (Reviewer #3).

The reviewers have discussed the reviews with one another and the Reviewing Editor has drafted this decision to help you prepare a revised submission.

Summary:

In this study, the authors recorded in the dorsal bank of the rhinal sulcus in rats engaged in a sucrose discrimination task. Rats discriminated between high and low intensity sucrose in order to respond correctly for a water reward. The authors report neural correlates in each area of sucrose intensity. Populations of neurons could be divided into those that were non-selective and a much smaller population that seemed to contain the majority of information about intensity. The authors used a novel probe test procedure to look at coding during generalization, which provided additional validation. Further they also reported on correlates with direction or response movement in OFC and aIC but not pIC. This dichotomy nicely adds to existing evidence that anterior regions are involved in behavior and decision making and provides an interesting contrast with posterior areas that I think is new and important.

Essential revisions:

The reviewers identified a number of concerns. Those deemed essential in discussion were that the authors clarify – tone down – their implied claims that there is something important about compressing activity from all three areas. This was mentioned in two reviews in particular. It seems hard to distinguish this from the benefit of just having larger ensembles, without real-time data. One option might be to do a controlled comparison of X cells from one area versus a similar number sampled across regions. Additionally, there was interest in more information about the potential confound between licking and the representation of intensity as well why the neurons were characterized as a small population.

Reviewer #1:

In this study, the authors recorded in the dorsal bank of the rhinal sulcus in rats engaged in a sucrose discrimination task. Rats discriminated between high and low intensity sucrose in order to respond correctly for a water reward. The authors report neural correlates in each area of sucrose intensity. Populations of neurons could be divided into those that were non-selective and a much smaller population that seemed to contain the majority of information about intensity. The authors used a novel probe test procedure to look at coding during generalization, which provided additional validation. Further they also reported on correlates with direction or response movement in OFC and aIC but not pIC. This dichotomy nicely adds to existing evidence that anterior regions are involved in behavior and decision making and provides an interesting contrast with posterior areas that I think is new and important. Overall the study was interesting, and well conducted. I have only a few points.

The first is whether the authors can decouple licking from the encoding. I think these are likely confounded and this group has previously reported that OFC activity and activity in other areas I think "represents" licking. I am wondering if anything further can be said about this relationship across the different regions here, independent of coding of intensity.

The second is the implications of the collapse across multiple brain regions in the decoding analyses. It seems to me that the authors imply that this provides some sort of meaningful result beyond what would be obtained simply by having more neurons in a given area. It seems to me that this would be a particularly novel result. But if I follow the methods correctly, each subject only contributed activity from one area. So I felt like this was overstated. I can see what the authors want to say, but I don't see that the data support it.

Otherwise it is a compelling report I think on sweetness coding in these areas.

Reviewer #2:

This manuscript represents an advance in our understanding for sucrose intensity coding in taste cortices. The authors employ a new, albeit simple, behavioral paradigm to define behavioral perception of variant sucrose intensities and the same task to determine how sucrose intensity and corresponding behaviors in the behavioral paradigm are encoded by neurons in both primary (insular cortex) and secondary (orbitofrontal) cortices.

The authors main finding is that a distributed, yet modest sized population of neurons through cortex represents sweetness intensity. This is novel and adds, as well as in some cases as the authors appropriately discuss, conflicts with, previous reports in manners which I expect may advance the field. The recordings and behavior are carefully executed as are the abundantly quantitative analyses. Setting aside minor issues which may be personal opinions on presentation of the results, I have only moderate comments for improvements to this otherwise compelling manuscript.

The assertion/conclusion that the intensity code for sucrose is distributed across as a "small population" of neurons I think warrants a reconsideration. The reference towards small is in relation to the population the authors managed to record from with electrode arrays. No doubt the number they acquired is notably large. However, in relation to total of neurons in these cortices (AI and OFC) the total population they would estimate encode taste intensity (assuming their recordings are unbiased in the sampled population) would in fact be very large.

The title is quite bold to imply that absolute decision making is occurring in taste cortices. I can see this based upon their data for the OFC neurons and outcomes. But not so much for the AI outcomes. Might the authors adjust wording to allow readers a more forthcoming appreciation for the role of these primary and secondary cortices in taste-related decision making?

*Reviewer #3:*

The manuscript by Fonseca et al. uses an elegant new novel behavioral task, combined with electrophysiological recordings in pIC, aIC and OFC, to describe how sucrose intensity is represented in these 3 cortical areas. The study addresses an important open question in the field using well-designed experiments. I have a few concerns regarding specific analyses and interpretation of results, but once these are addressed using additional analyses and changes to the text, this should be a valuable contribution to the field.

Concerns regarding analyses:

1) Cue-D high vs. low:

Because the duration of licking is different between these two conditions, comparison between 'high' and 'low' should only be at times either with or without licking separately across 'high' and 'low'. This is because licking itself affects activity of these neurons. This might be especially true for "Coh. neurons". For example, in Figure 2, bottom OFC 'Coh-Inact' neuron, the difference between 'high' and 'low' seems to stem from differential activity between licking and non-licking periods, regardless of concentration. Therefore, the data should be reanalyzed while comparing only periods with or without licking separately.

2) Spike timing:

a) The fact that statistically 'non-selective' neurons could be used to decode low/high does not necessarily suggest that spike timing is actually used by downstream neurons to decode information. It could simply be that some non-selective neurons have subtle differences in firing rates between high/low that are not consistent enough across trials for single neuron statistical tests. However, when many neurons are pooled together for decoding they are sufficient for decoding. Therefore, I suggest the authors either mention this as an alternative interpretation, or just remove this interpretation from the text.

b) The shuffling procedure used to eliminate spike timing will also eliminate inter-neuronal correlations (e.g., noise correlations) which are considered to be potentially useful for decoding. To test if this is the case, the authors can identify those pairs/groups of neurons with high correlations across trials and shuffle their spike times together (or any other procedure that will keep the pairwise correlation distributions the same), thereby eliminating the timing information while preserving inter-neuronal correlations.

3) Enhanced decoding of information when pooling across the 3 recorded regions:

As the authors themselves mention in the Discussion section, this enhancement could be a result of just having more neurons. For example, if more electrodes (or newer high density ephyz probes like neurpixels) were used in each region to yield more neurons per region, perhaps one region would have been sufficient for a level of decoding that is similar to behavior. Thus, if this result merely reflects a technical limitation of the recording technique, it does not really enable the reader to draw any useful conclusions. Alternatively, perhaps there is something about combining activity patterns across these different brain regions (each having a somewhat different complement of neuronal response types, as demonstrated by the authors) that facilitates decoding. The authors can test this with further analyses. For example, the authors can artificially inflate the population in each region by randomly choosing neurons and duplicating them to yield a number of neurons per region that is similar to the total across the 3 regions, and then compare the decoding.

Concerns regarding interpretation and discussion of results:

1) The authors repeatedly state that all 3 recorded regions represent sucrose intensity. However, this statement does not really present a particular insight regarding what might actually be represented in each brain region. Populations of concentration selective neurons in these 3 brain regions do not necessarily represent the same information. Based on the literature, pIC as the potential "primary gustatory cortex" could represent the sensory aspect of different gustatory sensory input. In contrast, OFC responses could represent subjective relative value (higher vs. lower sucrose concentration should have a relatively higher value). Interpretation of the representations in aIC remains more speculative, but would still be useful in the context potentially being an intermediate palatability representation between pIC and OFC. I suggest the authors add some discussion/speculation of what these concentration-selective responses might represent in these different regions.

2) The large fraction of non-selective responses to sucrose in all regions could reflect the fact that thirsty rats are getting water from the sucrose cue. This would imply that all regions have broad representations of need-relevant reward.

---

## [Author Response]

Essential revisions:The reviewers identified a number of concerns. Those deemed essential in discussion were that the authors clarify – tone down – their implied claims that there is something important about compressing activity from all three areas. This was mentioned in two reviews in particular. It seems hard to distinguish this from the benefit of just having larger ensembles, without real-time data. One option might be to do a controlled comparison of X cells from one area versus a similar number sampled across regions. Additionally, there was interest in more information about the potential confound between licking and the representation of intensity as well why the neurons were characterized as a small population.

Below are our point by point replies to all the reviewers comments. To address some of their concerns we added four supplementary figures (Figure 1—figure supplement 2, Figure 2—figure supplement 2, Figure 3—figure supplement 2, Figure 3—figure supplement 3) and, as requested, changed the colormaps of two figures (Figure 3 and Figure 3—figure supplement 1) from jet to inferno, since this color scheme is visible to most color blind people in a way that the colors are uniformly perceived (as suggested by a comment from BioRxiv). We have also toned-down the claim of compressing activity from all three areas, since after performing the analysis suggested by the Editor and reviewer 3, we found that the improvement in population decoding after combining activity across regions was mostly due to having larger ensembles rather than to a specific contribution from a particular area. Thus, we agree with reviewers about the importance of this point, and accordingly, we have decided to remove it.

Furthermore, we have now characterized in fine-grain detail the potential confound between licking and the representation of sucrose intensity that is plotted in Figure 2 with the corresponding PSTHs of licking behaviors. As we noted in the original version, rats licked shorter times for Low than High sucrose concentrations. Nevertheless, we believe that is unlikely that the difference in licking entirely explains the representation of sucrose intensity. This is because 45.1% of all Intensity-selective neurons have a best-window with no differences in licking (see Figure 2). The remaining 54.9% of neurons have a lick rate difference inside the best-window but most frequently they only covered a small fraction of the window (Figure 2—figure supplement 2). This facet can be seen in Figure 2 (grey-line above the PSTHs). Specifically, the overlap of the lick rate differences covered 31.4% of the entire best-window (Figure 2—figure supplement 2). This information is now included in the Materials and methods, and the Results section. Furthermore, we analyzed the relationship between licking and task performance and found that rats lick more rhythmically and similarly for both cues in sessions where their performance was better. This is reflected by a positive correlation between Low and High licking PSTHs and task performance (r=0.17 p<0.003; see Figure 1—figure supplement 2). Thus, rats did not solve the task by licking differently for both Low and High cues. We have previously found a similar result in a Go/NoGo taste discrimination task (Gutierrez et al., 2010), suggesting that for rats this seems to be a general behavioral strategy. For all these reasons, we conclude that is unlikely that most sucrose intensity representation is merely attributed to differences in licking behavior.

As suggested by reviewer 1, we also performed additional analysis in lick-coherence beyond intensity coding and found that in the three brain regions the coherence values selectively increased during the Stimulus epoch in comparison to pre-Stimulus and Outcome epochs. This result suggests that coherence between licks and spikes does not only reflect oromotor responses, rather it gates sensory information processing. This information is included in Figure 3—figure supplement 3 and the Discussion section.

We have also clarified that Intensity-selective neurons is a “small” population relative to non-selective neurons. Nevertheless, 18% of neurons in the rat’s brain could in fact be a larger population. Finally, and as suggested by reviewer 3, we also measured noise-correlation (between pairs of simultaneously recorded neurons from different channels) and found that it did not significantly contribute to the population decoding of intensity; thereby, demonstrating that is spike timing (and not noise-correlation) that conveys the most information about sucrose intensity.

We hope that these changes addressed all concerns rise by reviewers, and the manuscript can be now accepted for publication.

Reviewer #1:[…] Overall the study was interesting, and well conducted. I have only a few points.The first is whether the authors can decouple licking from the encoding. I think these are likely confounded and this group has previously reported that OFC activity and activity in other areas I think "represents" licking. I am wondering if anything further can be said about this relationship across the different regions here, independent of coding of intensity.

We agree with reviewer 1 about the importance in this point, and as we noted in the manuscript rats licked for slightly shorter times for Low compared to High concentrations (Figure 1E). Based on this result, and to reduce the impact of licking upon neural activity, we restricted our analysis from 0 to 0.6 s after cue delivery, since this time corresponds to the mean latency to stop licking after Low sucrose delivery (see Figure 1E). Nevertheless, we acknowledge that although licking is a highly stereotypic behavior, idiosyncratic and different lick patterns could still be displayed by different rats (see Figure 2 dashed PSTHs). For this reason, we have now further characterized in greater detail the contribution of licking upon the neural representation of sucrose intensity. First, for each session, we identified the times where rats licked significantly different for both cues (Figure 2 dashed lines and gray lines on top). Then, from all Intensity-selective neurons, we quantified how many best-windows contained no difference in licking behavior. We found that 45.1% of Intensity-selective neurons had a best-window with no difference in licking. For the other 54.9% of the neurons, we further quantified the amount of overlap between the lick differences and the best-window (Figure 2—figure supplement 2). On average, the lick differences only covered less than one third (31.4%) of the entire best-window (Figure 2—figure supplement 2), indicating that the lick rate differences could not entirely explain the neural representation of sucrose intensity. These results are now in the Results and Discussion section.

Regarding “if we have something else to say about lick coherence between regions, independently of intensity coding”. We first explored if the coherence values of all significant Lick-coherent neurons were different among regions. We found that the coherence values of the OFC (0.24 ± 0.005) were significantly lower relative to pIC (0.26 ± 0.003) and aIC (0.26 ± 0.003) (F_(2, 1672)_=3.772; p=0.02) (Figure 3—figure supplement 3A). Therefore, the pIC and aIC had not only a higher proportion of Lickcoherent neurons than OFC, but also IC neurons were better entrained with rhythmic licking. More importantly, we also uncovered, for the first time, that the level of coherence was significantly higher in the Stimulus-epoch in comparison with the preStimulus and the Outcome epochs. This result reinforces the idea that lick coherence does not just reflect oromotor responses, rather it is also involved in gating the input of sensory and taste information and perhaps in coordinating communication across brain regions. This point is now in the manuscript (Results and Discussion) and Figure 3—figure supplement 3B.

The second is the implications of the collapse across multiple brain regions in the decoding analyses. It seems to me that the authors imply that this provides some sort of meaningful result beyond what would be obtained simply by having more neurons in a given area. It seems to me that this would be a particularly novel result. But if I follow the methods correctly, each subject only contributed activity from one area. So I felt like this was overstated. I can see what the authors want to say, but I don't see that the data support it.

First, thank you for this insightful comment. To address this issue, for each region, and as suggested by reviewers, we matched the number of neurons for each population (Non-evoked, All, Non-selective, Intensity-selective) to exactly match the number recorded combining the three brain regions (pIC+aIC+OFC), this was achieved by using sampling with replacement. In this manner, each population of neurons of each single brain region (see Author response image 1) matched the result of collapsing the three structures (see panel B). The results confirmed what the reviewers were concerned about, namely that the improvement in the decoding accuracy when the three brain regions were collapsed was due to an increase in the number of Intensity-selective neurons. Therefore, we removed this point from the paper.

**Author response image 1. respfig1:** Decoding accuracy when each population (Non-evoked), All, Nonselective, and Int-selective) of each brain region matches the number of neurons sampled across regions collapsed (pIC+aIC+OFC). (**A**) Decoding accuracy when the number of neurons in a population (Non-evo, All, Non-sel or Int-sel) matched the number sampled from the three structures combined. That is, the number of neurons in each population of a brain region was the same as in the three regions combined (pIC+aIC+OFC: nNon-evoked = 200, nAll = 3527, nNon-selective = 2701, and nInten-selective = 596). It is seen that decoding accuracy obtained from each population of each region was not significantly different from the result obtained by combining the three structures together (panel B). Note that decoding accuracy obtained by using Non-evoked and Non-selective population never reached that of Intensity-selective neurons. * Significantly different relative to Non-evoked neurons from the same structure. # In comparison to the same population of the three regions combined.

Reviewer #2:[…] Setting aside minor issues which may be personal opinions on presentation of the results, I have only moderate comments for improvements to this otherwise compelling manuscript.The assertion/conclusion that the intensity code for sucrose is distributed across as a "small population" of neurons I think warrants a reconsideration. The reference towards small is in relation to the population the authors managed to record from with electrode arrays. No doubt the number they acquired is notably large. However, in relation to total of neurons in these cortices (AI and OFC) the total population they would estimate encode taste intensity (assuming their recordings are unbiased in the sampled population) would in fact be very large.

We agree the total number of Intensity-selective neurons in the whole region might be large in fact. To avoid further misunderstandings, we clarified in the Discussion section that “Although intensity-selective neurons comprise a “small” population relative to Non-selective neurons (Figure 3B and Table 2), we note that 18% of neurons in a rat’s cortex would represent a large population.”

The title is quite bold to imply that absolute decision making is occurring in taste cortices. I can see this based upon their data for the OFC neurons and outcomes. But not so much for the AI outcomes. Might the authors adjust wording to allow readers a more forthcoming appreciation for the role of these primary and secondary cortices in taste-related decision making?

After serious consideration of this point we decided to preserve a general title that encapsulates the big picture that represents the data in the article. However, we did take into consideration to add the animal species used. Furthermore, we also added a paragraph at the Discussion, suggesting what each brain region could be contribute in taste and decision making.

Reviewer #3:[…] Concerns regarding analyses:1) Cue-D high vs. low:Because the duration of licking is different between these two conditions, comparison between 'high' and 'low' should only be at times either with or without licking separately across 'high' and 'low'. This is because licking itself affects activity of these neurons. This might be especially true for "Coh. neurons". For example, in Figure 2, bottom OFC 'Coh-Inact' neuron, the difference between 'high' and 'low' seems to stem from differential activity between licking and non-licking periods, regardless of concentration. Therefore, the data should be reanalyzed while comparing only periods with or without licking separately.

As noted in the response to reviewer 1 (answer 1), we have now shown the PSTHs for licking in Figure 2 and quantify in more detail the times where licking was different for Low or High cues (see answer to reviewer 1 and Figure 2—figure supplement 2). Based on these results, we proposed that licking differences cannot entirely account for all neuronal responses evoked by sucrose concentration.

2) Spike timing:a) The fact that statistically 'non-selective' neurons could be used to decode low/high does not necessarily suggest that spike timing is actually used by downstream neurons to decode information. It could simply be that some non-selective neurons have subtle differences in firing rates between high/low that are not consistent enough across trials for single neuron statistical tests. However, when many neurons are pooled together for decoding they are sufficient for decoding. Therefore, I suggest the authors either mention this as an alternative interpretation, or just remove this interpretation from the text.

No doubt this is a possibility, accordingly we have now included in the Results section, the following sentence:

Specifically, we wrote, “Interestingly, the Non-selective group also decoded sucrose intensity significantly above chance level. […] Alternatively and despite their similar firing rates (spike counts) evoked by Low and High Cue-D, these neurons could use spike timing to discriminate sucrose concentrations (DiLorenzo and Victor, 2013).”

b) The shuffling procedure used to eliminate spike timing will also eliminate inter-neuronal correlations (e.g., noise correlations) which are considered to be potentially useful for decoding. To test if this is the case, the authors can identify those pairs/groups of neurons with high correlations across trials and shuffle their spike times together (or any other procedure that will keep the pairwise correlation distributions the same), thereby eliminating the timing information while preserving inter-neuronal correlations.

We thank the reviewer for this thoughtful comment. We acknowledge that shuffling spikes can also affect noise-correlations. Since, it is known that spikes in a pair of neurons simultaneously recorded can covary across the session, a phenomenon known as noise-correlation. Moreover, noise-correlations are thought to vary with attentional, behavioral, and overall brain-state of the network, but the function of this phenomenon is not completely understood. Nevertheless, it is well known that noise-correlation could affect (either increase or decrease) population decoding (Carnevale et al., 2013; Zohary et al., 1994). Therefore, we determined the impact of removing noise correlation over the decoding accuracy of sucrose intensity and found that in our experiments noise-correlation appears to have no significant effect over decoder accuracy. Thus, the drop in decoding accuracy after spike timing-shuffling was not due to noise correlation.

We now briefly describe more details of how we did the analysis. First, we could not shuffle together the spikes of neuron pairs with noise-correlation, as suggested by reviewer 3, because one neuron could also be correlated to more than one neuron, that is not necessarily correlated between each other. Furthermore, shuffling together the spikes of neurons that are not correlated between them but correlated to a common neuron, might introduce noise-correlation within this pair of neurons that were not correlated from the beginning. Therefore, we decided to implement a different approach to abolish noise-correlation and then measure its contribution for population decoding. First, we identified the pairs of correlated neurons that were recorded in the same session but in different channels. Then for each neuron we calculated the firing rate during different windows of the Stimulus epoch, from 0.1 to 0.6 s in 0.1 s steps, and normalized this activity relative to each trial type (see Materials and methods section) (Ecker et al., 2010; Carnevale et al., 2015). Then, we computed the Pearson correlation coefficient of normalized spike counts (noise correlation, named r_sc_) from Neuron A and Neuron B. If the r_sc_ was significant (see Figure 3—figure supplement 2A, upper panel), then we shuffled the trials positions of neuron A and those for neuron B to remove noise-correlation (see Figure 3—figure supplement 2A, lower panel) (Averbeck et al., 2006; Cohen and Kohn, 2011). This procedure was repeated 10,000 times in order to obtain a corrected *p*-value. If correlation was still significant then each neuron was designated as a noise correlated pair of neurons. Subsequently, we shuffled trials positions (named shuffled-trials) of neurons with noise-correlation and the decoder was fed with these shuffled-trials matrices and unshuffled matrices from non-noise correlated neurons of the Non-evoked, All, Non-selective, and the Intensity-selective population. Methods are described in detail in the Materials and methods subsection- “Population decoding of sucrose intensity” and the results are plotted in Figure 3B and Figure 3—figure supplement 2. Results show that shuffled trials effectively remove noise correlation (see Figure 3—figure supplement 2B), yet it did not affect the decoding accuracy (Figure 3B). Thus, in our experiments noise-correlation appears to have no significant effects over decoding accuracy. Therefore, the impairment in decoding accuracy observed after spike timing-shuffling was not due to noise correlation.

3) Enhanced decoding of information when pooling across the 3 recorded regions:As the authors themselves mention in the Discussion section, this enhancement could be a result of just having more neurons. For example, if more electrodes (or newer high density ephyz probes like neurpixels) were used in each region to yield more neurons per region, perhaps one region would have been sufficient for a level of decoding that is similar to behavior. Thus, if this result merely reflects a technical limitation of the recording technique, it does not really enable the reader to draw any useful conclusions. Alternatively, perhaps there is something about combining activity patterns across these different brain regions (each having a somewhat different complement of neuronal response types, as demonstrated by the authors) that facilitates decoding. The authors can test this with further analyses. For example, the authors can artificially inflate the population in each region by randomly choosing neurons and duplicating them to yield a number of neurons per region that is similar to the total across the 3 regions, and then compare the decoding.

We completely agree with the reviewer 3 and as noted in answer to reviewer 1, we have decided to remove this point from the manuscript.

Concerns regarding interpretation and discussion of results:1) The authors repeatedly state that all 3 recorded regions represent sucrose intensity. However, this statement does not really present a particular insight regarding what might actually be represented in each brain region. Populations of concentration selective neurons in these 3 brain regions do not necessarily represent the same information. Based on the literature, pIC as the potential "primary gustatory cortex" could represent the sensory aspect of different gustatory sensory input. In contrast, OFC responses could represent subjective relative value (higher vs. lower sucrose concentration should have a relatively higher value). Interpretation of the representations in aIC remains more speculative, but would still be useful in the context potentially being an intermediate palatability representation between pIC and OFC. I suggest the authors add some discussion/speculation of what these concentration-selective responses might represent in these different regions.2) The large fraction of non-selective responses to sucrose in all regions could reflect the fact that thirsty rats are getting water from the sucrose cue. This would imply that all regions have broad representations of need-relevant reward.

We completely agree. It is possible that each brain region encodes different features of the sucrose intensity cues. This possibility is now clearly highlighted in the Discussion section.

Specifically, we have now written:

“However, the fact that all three areas decoded sucrose intensity equally well does not imply that they represent the same information, but rather they may encode different features of the sucrose intensity cues. […] Finally, the OFC responses during the Stimulus epoch could also signal the relative reward value of Low and High sucrose cues (Rolls et al., 1990; Tremblay and Schultz, 1999).”